# A Review of Ecological Assets and Ecological Products Supply: Implications for the Karst Rocky Desertification Control

**DOI:** 10.3390/ijerph191610168

**Published:** 2022-08-17

**Authors:** Jiayi Zhou, Kangning Xiong, Qi Wang, Jiuhan Tang, Li Lin

**Affiliations:** School of Karst Science, State Engineering Technology Institute for Karst Desertification Control, Guizhou Normal University, Guiyang 550001, China

**Keywords:** ecological assets, ecological products, supply capacity, progress, implications, karst rocky desertification control

## Abstract

Ecological assets refer to natural resource assets that can provide ecological products and services for human beings. Researching ecological assets and the supply of ecological products contributes to the sustainable development of natural–social–economic complex systems. This study conducts a literature search and statistical analysis based on the Web of Science, CNKI and Foreign Journal Resource Service System of Guizhou Normal University Library literature databases. We review 117 publications on the studies of ecological assets and ecological products supply. Based on summarizing the landmark results, the key scientific issues that need to be solved are proposed, and their implications for karst rocky desertification control are discussed. The results show that: (1) the number of publications each year from 2001 to 2020 shows a fluctuating growth trend. (2) The research concentrates on four dimensions: theoretical, evaluation, mechanism, and strategies research. Among them, evaluation research is the focus and hotspot of the research. (3) It is necessary to expand the research on the definition and distinction of ecological assets, services and products; the evaluation system of ecological products; the supply mechanism of ecological products from the perspective of resource endowment and the research of ecosystem coherence at different scales. (4) The karst rocky desertification control should focus on ecological assets’ management to promote the supply capacity of ecological products, pay attention to the quality assessment of ecological assets, explore the influencing mechanism of ecological assets and its optimization and promotion paths and strengthen the research on village ecosystems under karst rocky desertification control.

## 1. Introduction

Ecosystems maintain the ecological environment on which human beings depend, provide various direct or indirect ecosystem services for human social development and are a necessary foundation to support the sustainable development of economy and society [1]. Since the 20th century, under the influence of population pressure and economic development, to meet the growing material and spiritual needs, the negative impact of humans on the ecosystem has appeared on a large scale. The Millennium Ecosystem Assessment pointed out that out of 24 evaluated ecosystem types, 15 are being damaged; about 60% of the ecosystem services on which human beings depend, such as drinking water supply, fisheries, regional climate regulation and natural disasters and pest control, continued to decline [2]. The degradation of ecosystem services jeopardizes the wellbeing of contemporary human society, and it will significantly reduce the benefits that human descendants can obtain from the ecosystem, causing an irreversible reversal of the human living environment [3]. Therefore, strengthening ecological construction and maintaining ecological security are the common themes faced by humanity in the 21st century.

Ecological assets are the value performance of natural resources, ecosystems and ecological environment with human economic and social development as the core [4], divided into stock and flow. The ecological assets stock refers to the ecosystem itself, such as forest and grassland ecosystems. The ecological assets flow refers to the product of the continuous output of the ecosystem, or the effect that continuously occurs due to its existence, that is, the supply, regulation and cultural services in the ecosystem [5], for example, forest product supply, water conservation services, leisure and tourism. The increment or flow of ecological resource assets based on the stock is ecological products [6]. The stock of ecological resources and the sustainability of the ecosystem are the sources of the supply of ecological products. From the perspective of resource endowment, the main body of ecological product supply is the area with abundant ecological resources; the object of supply is the area with insufficient ecological resources; and the essence of enhancing the supply capacity of ecological products is to maintain, improve and expand the stock of ecological resources, enhance the regional ecosystem service functions and enhance the ecosystem’s sustainability issues [7].

Ecological environment problems in ecologically fragile areas have always been given attention. Karst refers to the dissolution of water on soluble rocks (carbonate rocks, etc.) and the resulting various landscapes and phenomena occurring on the surface and underground [8,9], which is an apparent binary landform morphological structure [10]. Karst landscapes are widely distributed worldwide, accounting for 15% of the world’s total land area [11,12,13]. Concentrated karsts are mainly distributed in central and southern Europe, eastern North America and southwestern China [14,15]. The southern China karst area is the most typical, complex landscape-type area in the world [16], called the world’s karst museum [17]. The karst area is restricted by the unique geological conditions of soluble carbonate rocks, the soil formation is prolonged, the soil layer is thin and discontinuous and the hydrological process responds rapidly [18]. Unscientific human activities lead to soil erosion in the karst environment, exposing the bedrock under the soil and forming a landscape similar to the desert on the surface [19,20,21,22,23]. This process is called karst rocky desertification [19,20,21,22,23]. Karst rocky desertification has caused problems, such as a decline in land productivity, sharp reduction in surface vegetation coverage, weakening of water system conservation capacity, depletion of surface water sources, loss of land resources and reduction in food production [24], resulting in the gradual loss of ecosystem services and threatening the sustainability of the region’s development. To this end, the Chinese government launched a national, comprehensive, rock desertification control program at the end of the 20th century [25]. Since then, the expansion of rock desertification, the destruction of vegetation and soil degradation have been initially alleviated, and phased results have been achieved [26,27]. However, there are still other problems, such as a difficulty in effectively maintaining the governance results, the inability to eliminate the contradiction between human beings and the land and an increasing difficulty in governance when dealing with rocky desertification in China [28]. How to scientifically promote the governance of karst rocky desertification, improve the quality and sustainability of ecological restoration and enhance the ability of ecosystems to provide ecological products and services have become critical issues for the harmonious co-existence of humans and nature in karst rocky desertification areas.

Ecological assets are part of the ecosystem. The management of ecological assets is an effective way to change the structure of the ecosystem, the key to changing the quality of the ecosystem and a way to improve the supply capacity of ecological products. Therefore, it is necessary to review the research progress of ecological assets and ecological product supply in order to guide the work of karst rocky desertification control. Scholars have conducted many studies on ecological assets and ecological products supply, but the progress and shortcomings of the current research on the ecological assets and ecological products supply are unknown. Therefore, this paper systematically reviews the research progress of ecological assets and ecological product supply, and summarizes the main research progress and landmark achievements in this field from four aspects: theoretical, evaluation, mechanism, and strategies research. Four key scientific issues to be solved are put forward; this points out the direction for further research on ecological assets and ecological products supply. Their implications for rocky desertification control are discussed, and scientific and technological references for promoting the sustainable development of rocky desertification control are provided.

## 2. Materials and Methods

To identify the relevant studies, a search was conducted based on certain platforms, including the Web of Science (WOS), Foreign Journal Resource Service System and China National Knowledge Infrastructure (CNKI). The time range for our retrieval ranged from 1 January 2000 to 31 March 2022. Figure 1 shows the process of the literature search and screening.

We finally obtained 47 English and 70 Chinese publications. The top 3 contributors in number of literature of the topic were Huiyi Yu (4), Baiming Chen (3) and Xiahui Wang (3). When calculating the number of published studies by authors, all authors in the literature were accounted for, regardless of the author’s order. The low number of articles published by each author presented the lack of systematic and in-depth research in the field.

We divided the research according to the subject of the article. If the article only differentiated between and analyzed related concepts, it was classified as theoretical research. If the whole article only included accounting (ecological asset accounting, ecosystem service value, ecological product value accounting or ecological product supply capacity assessment), it was classified as evaluation research; if the article analyzed ecological assets or ecosystem service impact factors, it was classified as mechanism research; and if the article conducted research on certain strategies, such as ecological compensation and ecological function zoning on the basis of accounting, it was classified as strategic research.

## 3. Results

### 3.1. Literature Statistical Analysis

#### 3.1.1. Annual Distribution

The research conducted on the ecological assets and supply of ecological products generally shows a trend of fluctuating growth. Figure 2 shows that the period between 2000 and 2022 can be roughly divided into two stages. In the first stage (2000–2010), the number of documents was generally low, and the annual number of documents was not more than five, in the starting stage. In the second stage (2011–2022), the number of documents increased at the stage of fluctuating growth.

#### 3.1.2. Content Distribution

The research content of all the literature were classified and summarized according to theoretical, evaluation, mechanism and strategies research. Among them, theoretical research literature accounted for 16%; evaluation research papers accounted for 4%; mechanism research papers accounted for 36% and strategies research literature accounted for 8%. The evaluation and mechanism research were the hot topics and focus of the present research.

#### 3.1.3. Institution Distribution

At present, there are many research institutions. Northeast Forestry University (six papers) ranked first, followed by the Institute of Geographical Sciences and Natural Resources Research, Chinese Academy of Sciences (five papers), Zhejiang Ocean University (three papers), Beijing Normal University (three papers), Chinese Academy of environmental planning (three papers), Beijing Forestry University (three papers), Chongqing University (three papers), Shenyang Institute of Applied Ecology, Chinese Academy of Sciences (three papers), Institute of Mountain Hazards and Environment, Chinese Academy of Sciences (three papers) and China University of Geosciences (three papers). Among all the research institutions, there were more foreign research institutions than domestic ones, but most of the research institutions ranked high in the number of studies for domestic institutions.

### 3.2. Main Progress and Landmark Achievements

#### 3.2.1. Theoretical Research

##### The Concept of Ecological Assets, Ecosystem Services and Ecological Products

The concept of ecological assets was developed from ecosystem services and natural capital proposed by foreign scholars, and the term ecological assets is more widely used in China. For the concept of ecological assets, the Chinese academic community has not reached a consensus. Throughout the development history of the concept of ecological assets (Table 1), it can be divided into two categories. One of those focuses on the value embodiment of ecological assets [4,29,30,31,32] and its representative view is that ecological assets include two systems of natural capital as “stock” and ecosystem services as “flow”. These combine natural resource and ecosystem service value, and should include all-natural resources and ecological environments that can provide services and welfare for human beings [4]. The other focuses on the physical form of ecological assets [33,34,35]; some of them emphasize ecological assets as ecological landscape entities, and some emphasize ecological assets as ecological economic resources. The representative view is that ecological assets are the main components of natural resource assets that can provide ecological products and services for human beings, including forests, shrubs, grasslands and wetlands and deserts, and other natural ecosystems, such as farmland, urban green spaces and other artificial ecosystems based on natural ecological processes, as well as wildlife resources [35].

Scarcity and property rights are the social attributes of ecological assets. Ecological assets come from ecological resources. Ecological resources are not equal to ecological assets. Ecological resources can be transformed into ecological assets. The transformation of ecological resources into ecological assets requires certain conditions: scarcity, clear property rights and the ability to generate income. Scarce ecological resources stimulate people’s desire to possess, and lead to changes in the owners of ecological resources rights and interests, thus transforming ecological resources into ecological assets. Therefore, ecological assets have scarcity and property rights. Generally speaking, most of the ecological assets in a physical form (such as fruits, timber, water resources, etc.) can determine ownership. Invisible ecological assets (such as air purification and oxygen supply, climate regulation, landscape enjoyment, etc.) are mostly public, and it is difficult for any organization or individual to determine their ownership [4].

Ecosystem services refer to the collection of various tangible and intangible products provided to human beings by the ecosystem, exerting its functions through the interaction process between humans and natural society [37]. MA divides ecosystem services into provision, support, regulation and cultural services. Provision services are the products obtained from ecosystems, including food, water, wood and fiber, etc. Regulation services are the benefits obtained from the regulation of ecosystem processes, including the regulation of climate, floods, disease, waste and water quality. Cultural services are the nonmaterial benefits people obtain from ecosystems through spiritual enrichment, cognitive development, reflection, recreation and aesthetic experiences. Support services are those that are necessary for the production of all other ecosystem services, for example, the production of atmospheric oxygen, soil formation and retention, nutrient cycling, water cycling and provision of habitat [38]. The development process of the concept of ecosystem services is shown in Table 2.

The difference between these definitions is whether to distinguish between the service generation process, service acquisition process and service final benefit. MA [38] and Daily [41] show that as long as humans are the beneficiaries, ecosystem processes or functions are services. For Boyd [43], services are directly consumed components (structure included), meaning indirect processes and functions are not ecosystem services. Wallace [44] suggests that ecosystem processes should not be considered as services. Ecosystem services are considered as benefits by Costanza [40], whereas ecosystem services are not considered as benefits by Fisher [45]. Consistently, ecosystem patterns, processes and functions underlie the generation of ecosystem services. Ecosystem functions and ecosystem services do not correspond one-to-one. Ecosystem services and products are interdependent. Human wellbeing requires ecosystem services.

Although ecological products have been the topic of much academic study, no unified definition of such products has emerged. The development process of the concept of ecological products is shown in Table 3. In a narrow sense, ecological products are purely natural and produced by biological processes, including fresh air, clean water and a pleasant climate. Other natural products supply regulation, support and cultural services, which all contribute to human welfare [48,49,50,51]. In a broad sense, ecological products are produced jointly by human beings and nature, which not only include the pure, natural production of ecosystem services, but also the supply of agricultural and forestry products produced by human beings [52,53,54,55,56]. Agroforestry products with attached human labor can be understood as the provisioning services of ecosystems in which humans interact with nature. Therefore, ecological products are a part of ecosystem services. Wild fruits are ecological products in a narrow sense, and fruits grown by farmers are ecological products in a broad sense, but both belong to ecosystem supply services. Whether they are produced by biology or by biology and human beings, these products are developed under the condition that as the increasingly prominent contradictions between economic development and ecology and resources increase, the people’s appeal for a better life emerges. The two can coexist without violating each other [53].

##### The Relationship between Ecological Assets and Ecological Products

From the perspective of the ecological process formed by ecological assets and products, ecosystem components and structures are the main components of the stock of ecological assets, and ecosystem services are the main components of the flow of ecological assets. When the ecosystem itself and ecosystem services are given ownership, they can enter the market for trading, that is, the ecological assets are converted into ecological products (Figure 3) [5].

The quantity and quality of ecological assets and products are directly dependent on the ecosystem’s components, structures, processes and functions. Their distribution and changes follow the principles of ecosystem ecology, biological reproduction and related resource and environmental constraints, as well as ecological balance, which is also affected by human activities and development and utilization [57]. Ecosystems are producers of services and assets, and humans are consumers of services and assets. The ecological products supply side includes the ecosystem itself and the interaction between humans and ecology [53,56,58,59]. Ecological assets emphasize clear property rights and scarcity. Some ecological products, such as clean air, good climate and other natural elements, do not belong to ecological assets because they are not scarce and it is challenging to confirm ownership. In contrast, other ecological assets, such as some supply and regulation services provided by ecosystems, are similar to ecological products [58].

#### 3.2.2. Evaluation Research

##### Evaluation of Ecological Products Supply Capability

Due to the lack of a unified understanding of the definition and classification of ecological products in existing studies, there are few studies regarding the evaluation of ecological product supply capacity. A systematic evaluation system for ecological product supply capacity has not yet been formed. There are two main approaches for evaluating the supply capacity of ecological products in existing studies: the material quality method and composite index method.

The material quality of ecological products is used, that is, the output of various products provided by the ecosystem within a certain period of time, the functional amount of ecological regulation services and the number of tourists used to characterize the supply capacity of ecological products. Pang estimated the supply capacity of ecological products in the Hulunbuir Reserve from 2000 to 2010 in water conservation, soil conservation, carbon fixation and oxygen release [60].

Based on the connotation of ecological products, the index system for evaluating the supply capacity of ecological products was constructed, and the entropy value method was used to determine the index weights and calculate the comprehensive index of the supply capacity. Peng constructed an ecological product supply capacity evaluation index, including 4 primary categories; 8 secondary categories; and 24 specific indicators, including ecological environment products, ecological material products, ecological cultural products and ecological space products, and evaluated the supply capacity of ecological products in Beijing, Tianjin and Hebei [61].

##### Evaluation of Ecological Products’ Values

Ecological products, in a narrow sense, are ecosystem services. Therefore, some scholars use the ecosystem services accounting system to assess ecological products [49,50,51,52,53,54,55,56,57]. In addition, some scholars use the gross ecosystem product (GEP) to characterize the value of ecological products [62,63,64]. In 2013, OuYang Zhiyun borrowed the concept of gross domestic product (GDP) for the concept of gross ecosystem product, that is, the total value of products and services provided by the ecosystem for human beings; it is composed of 17 indicators of 3 functions: product provision, regulation function and cultural function [65]. This definition does not include support services, which is similar to the concept of ecological products. Therefore, some scholars use this method to study the value of ecological products.

There are two main types of ecological product value evaluation methods: the equivalent factor evaluation and functional value evaluation methods. The equivalence factor evaluation method constructs the value equivalents of various service functions of different ecosystems based on the quantifiable standards that distinguish between different types of ecosystem service functions, and then evaluates them in combination with the distribution area of ecosystems [66]. This method is based on the Millennium Ecological Assessment classification system and the classification method of Costanza et al. [39]. It uses the equivalent factor table of ecological service value per unit area studied by Xie Gaodi for the calculation. Due to the substantial spatial heterogeneity of ecosystem services, different scholars have corrected the ecological service value equivalent factor in specific applications from different perspectives, and two correction methods are as follows: first, make the regional revision according to the grain yield [67], and second, choose the corresponding indicators for revision, including, mainly, ecological vulnerability [68], ecological location [69], biomass [66], NPP, rainfall, habitat quality, soil erosion and accessibility regulators [70]. The functional value evaluation method is used to obtain the total value based on the number of ecosystem services and unit price of the available amount. Usually, the direct, alternative and simulated market methods are used for value transformations [66,71].

The market value method can be applied in an number of ways using actual expenses, opportunity costs, preventive and recovery costs, replacement costs, travel costs, hedonic price and contingent value calculations. The market value method refers to a method of valuing ecosystem products and functions with market prices, and is mainly used for the evaluation of material products produced by ecosystems. From the perspective of consumers, the expense method takes the sum of the relevant expenses actually paid by tourists for tourism as the tourism economic value of scenic resources. Opportunity cost method means that the use of a scheme under the condition of scarcity of resources means that other schemes must be abandoned, and the maximum benefit that may be obtained in the abandoned scheme constitutes the opportunity cost of the scheme. Assuming that a project with a similar actual effect but is not actually conducted is adopted, the shadow project method of replacing the economic loss of the project is assessed with the construction cost of the project. The recovery and preventive approaches assess how much people are willing to spend to avoid damage to ecosystem services or, if damaged, to restore them to their original levels. The replacement cost method presents how some ecosystem services can be replaced by human-made systems, and the value of these ecosystem services can be assessed by the cost of replacing the service. The travel cost method assesses the willingness of individuals to pay an equivalent travel expense in order to obtain the consumption of a non-market item. The hedonic price method estimates the value of environmental quality by assessing the price people pay for the enjoyment of a good environment. The contingent value method can be used to discover how people price certain environmental changes by interviewing the survey directly to the relevant population, the respondents expressing their demand for a specific environmental product or service. The advantages and disadvantages of each method are presented in Figure 4.

#### 3.2.3. Mechanism Research

The research conducted on the influencing factors of ecosystem services can help to accurately simulate future ecosystem service scenarios, and then serve the scientific formulation of strategies for improving the supply capacity of ecological products. To this end, many scholars have explored the effects of natural factors (climate change, soil, topography, vegetation) [72,73,74], socioeconomic factors (population, GDP, income, education, policy) [75,76] and human activities (land use, urbanization, ecological construction) [77,78,79,80,81,82,83,84,85,86,87,88,89,90,91] on ecosystem services. Land-use change can affect ecosystem service changes. The uneven, regional distribution and diversification of social and economic factors lead to a preference for different types of ecosystem services, leading to the difference in ecosystem service demand [92]. Since the changes in ecosystem services are often driven by the synergy of natural factors and human activities [93], regression models and correlation analyses are often used to separate their impacts on ecosystem services.

##### Impact of Natural Factors on Ecosystem Services

Natural factors, such as climate, soil, topography and biology, are the essential components of ecosystems and geographical units. Ecosystem structures and geospatial patterns affect ecological processes, determining the spatiotemporal distribution of ecosystem services [92]. Climate change is an essential factor affecting the supply of ecosystem services. On the one hand, climate change affects the supply level of ecosystem services, such as water supply, carbon sink, food supply, net primary productivity, vegetation coverage and biological distribution, by directly changing precipitation and temperature levels [94,95,96]. On the other hand, climate change affects the supply level of ecosystem services by changing land use/cover, ecosystem structure and composition, and resource-use intensity [87]. Topographic factors control the distribution of spatial hydrothermal resources, which affect many environmental conditions and ecological processes, such as actual solar radiation, temperature, soil mineralization rate and vegetation distribution, and directly determine the supply and maintenance of ecosystem services [92,97]. Li showed that the factors, such as increased precipitation, potash fertilizers and decreased distance from water sources, all had positive effects on improving Hulunbuir grassland ecosystem services [72]. Han and Dong believe that precipitation, evaporation and root depth are the main factors affecting the spatial pattern of water supply in Guizhou Province [73]. The spatial heterogeneity of soil consolidation is caused by rainfall erosivity, topography, cover management factors and support practices of land-use types [73].

##### Impact of Socioeconomic Factors on Ecosystem Services

The uneven regional distribution and diversified development of social and economic factors lead to a preference for different types of ecosystem services, which in turn leads to the trade-off and demand for different ecosystem services [82]. The economic development model employed by local officials or planners affects the accumulation and utilization of ecosystem service value by affecting land use, which in turn affects the per capita economic income and welfare obtained from ecosystem services. Optimizing the mode of economic development and reducing the interference and impact of human activities can enhance the service value of regional ecosystems and promote the coordinated development of the ecological environment and social economy [80]. Zhang showed that the supply of multi-ecosystem services was negatively correlated with population density, employed population and real GDP per capita, and had a significant positive correlation with the total fiscal revenue and number of teachers [81]. Han believed there was a weak, negative correlation between urbanization rate, per capita GDP, population density and ecosystem services [74].

##### Impact of Human Activities on Ecosystem Services

The influence of human activity on ecosystem services is highly complex. One type of human activity can affect multiple ecosystem services; many kinds of human activities can affect one ecosystem service [98]. Human activities, such as land-use, urbanization and ecological conservation projects, can change landscape patterns and ecological processes [71]. Human activities affect the ability of ecosystems to maintain biodiversity and provide ecosystem products by changing the Earth’s habitat, ecosystem structure and biogeochemical cycles [98]. Land use plays a decisive role in maintaining the ecological environment system. Land-use change directly affects the composition and morphology of the ecosystem, and ultimately affects the ability of the ecosystem to provide ecosystem services [78]. Urban expansion changes the structure of many ecosystems by changing land-use patterns [80,81]. Zhang showed that urban expansion transforms natural and semi-natural land cover into urban land (cultivated land into an artificial surface), which has a powerful, negative impact on the ecosystem service value of urban agglomerations along the coast of the Bohai Rim, China [95]. Ecological restoration projects can lead to significant changes in land-use and land-cover changes, and have been recognized by the international community as an essential means to enhance and maintain biodiversity and ecosystem services [99]. Gao and Zhang believed that karst rocky desertification governance ecological restoration projects have improved ecosystem services [83,84]

#### 3.2.4. Strategies Research

##### Improving the Supply Capacity of Ecological Products by Performing Ecological Function Zoning

Ecological function zoning is an essential part of ecosystem management. Its purpose is to conduct targeted protection according to the provision of ecological services, which can fundamentally curb the degradation of the ecological environment, maintain ecosystem balance, improve ecosystem services and improve the supply capacity of ecological products [100,101,102].

##### Improving the Supply Capacity of Ecological Products by Formulating Ecological Compensation Policies and Measures

Through the assessment and evaluation of ecological assets to understand the quality of ecosystems and their changes, combined with the spatial differences in the ecological product supply capacity, ecological compensation standards are determined and ecological compensation policies and measures are formulated so as to improve the ecological product supply capacity [103,104].

## 4. Discussion

### 4.1. Key Scientific Issues to Be Solved

#### 4.1.1. In View of the Unclear Concepts of Ecological Assets, Ecosystem Services and Ecological Products, the Concepts Should Be Defined Based on the Consideration of Measurability and Quantifiability

A clear definition of the concepts of ecological assets, ecological products and ecosystem services is the basic point and theoretical premise of the current research. There are many ecological products, considerable differences in characteristics and various forms of contributions to wellbeing. At the same time, some ecological products have been fully integrated into social and economic systems, overlapping with primary, secondary and tertiary industries. As a result, it is not easy to unify the understanding of ecological products in academic circles, and the relationships and boundaries concerning related concepts, such as ecosystem services, are not clear [105], which limits the in-depth research on the evaluation system of ecological product value and supply capacity, path mechanisms and policy guarantees for the realization of ecological product values. When people study ecological products, they need to define and explain the scope of ecological products in advance. In fact, which scope is allowed depends on the nature of the research topic and the problem to be solved [36]. In the future, the concept’s definition should be presented on the basis of considering the measurability and quantification of ecological assets, ecosystem services and ecological products.

#### 4.1.2. Given the Lack of Authoritative Ecological Product Evaluation Systems, Differentiated Evaluation Method Systems for Different Spatial Scales Should Be Established According to the Spatial Scale Characteristics of Ecological Products

Value accounting is a quantitative expression for the value function of ecological products, and is a prerequisite for the development, trading and compensation of ecological products [106]. Standardized pricing is one of the problems in realizing the value of ecological products. Applying a ‘price tag’ to system services and establishing a statistical evaluation index system for ecological products can effectively solve this problem [107]. However, due to different understandings of the connotation and characteristics of ecological products, existing studies lack a unified understanding of the definition and classification of ecological products; there are relatively few evaluations of ecological product supply capacity, and a systematic system of ecological product supply capacity evaluations and value accounting methods has yet to be developed. The accounting results lack the support of authoritative data, and their credibility and recognition are greatly affected. It is not sufficient to support the development of localized strategies to improve the supply capacity of ecological products and the choice of value-realization paths for the differentiation of ecological product types. Therefore, it is urgent to establish differentiated evaluation method systems for different spatial scales according to the characteristics of ecological products at different spatial scales.

#### 4.1.3. In View of the Unclear Supply Mechanism of Ecological Products from the Perspective of Resource Endowment, the Analysis of the Influencing Factors of Ecological Assets Should Be Strengthened

The change mechanism of ecological product supply capacity is the theoretical basis for formulating ecological product supply capacity improvement strategies. The supply capacity of ecological products depends on the natural ecosystem itself and human or human-led market entities [108]. Therefore, to improve the supply capacity of ecological products, it is necessary to combine natural resources and play the role of government and the market [48]. However, most of the existing research on improving the supply capacity of ecological products focuses on social entities, and the research on the ecological factor endowment is insufficient. The research on improving the ecological product supply capacity from the perspective of resource endowment is essentially the preservation and appreciation of ecological assets. The research conducted from the perspective of resource endowment should be strengthened in the future. On the basis of clarifying the relationship between ecological assets and product supply, the driving force of ecological assets should be revealed, and the supply capacity of ecological products should be improved by optimizing and enhancing ecological assets.

#### 4.1.4. Given the Lack of Studies on Ecosystem Coherence at Different Scales, Studies Conducted on Ecosystem Coherence at City–County–Township–Village and Smaller Scales Should Be Conducted on the Basis of Paying Attention to the Regional Differences

Strengthening the research on ecosystem coherence at different scales can effectively link ecological policy measures at all levels. Ecosystem service functions depend on ecological and geographic system processes on different spatial and temporal scales, and have strong spatial-scale characteristics [109,110]. There are differences in the difficulty of obtaining data at various scales and the scope of ecological function, in the selection of functional value types and the composition of ecological product values [106]. Existing studies on the ecological assets and ecological products supply mainly focus on single, largescale studies, such as regions, cities and counties, often neglecting the spatial variability within regions, making the effective articulation and guidance of spatial planning scale systems at various levels unattainable and measures insufficiently targeted. In the future, we should consider the strong spatial-scale characteristics of ecosystem services, pay attention to the spatial variability within regions and conduct studies on ecosystem coherence at city–county–township–village and smaller scales.

### 4.2. Implications for the Karst Rocky Desertification Control

The problem of karst rocky desertification lies in the loss of surface soil, exposed bedrock, loss of land productivity and degradation of the ecological environment caused by soil erosion [111]. The fundamental cause of karst rocky desertification is that humans ignore the ecosystem’s intangible services, care more about the specific products provided by the ecosystem and perform non-renewable activities. Rocky desertification control projects should increase the area of the ecosystem, improve the quality of the ecosystem and improve the ability of the ecosystem to provide ecological products and services. Based on the above analysis, there are mainly the following implications for rocky desertification control.

#### 4.2.1. Focus on Ecological Assets Management to Promote the Supply Capacity of Ecological Products

At the end of the 20th century, the Chinese government launched a national, comprehensive rocky desertification control program [25], forming ecosystems under rocky desertification controls in which human beings and nature interact. The ecosystems under rocky desertification control undertake critical ecological functions, such as water conservation, soil and water conservation and biodiversity maintenance, in karst areas. However, the ecosystems under rocky desertification control have a simple structure, poor stability and weak resistance, and the control results are difficult to maintain [112,113,114], which profoundly affects the supply capacity of ecosystem services and products. Regarding the definition of ecological assets, the author agrees with Ouyang Zhiyun’s point of view that ecological assets are all kinds of ecosystems and wildlife resources. Therefore, in rocky desertification control areas, the ecosystems under rocky desertification control can be regarded as ecological assets. The ecosystem produces supply, regulation, support and cultural services through ecological processes. Ecological products should include the total of ecological material products, ecological regulation services and ecological cultural services obtained by human beings from the ecosystems under rocky desertification control. Taking the rocky desertification control forest ecosystem as an example, the ecological products obtained by human beings include: prickly pears, plums, walnuts, loquats, prickly ash and other ecological material products; ecological regulation services, such as water conservation, carbon fixation and oxygen release and water purification; and ecological cultural services, such as leisure and tourism [115].

Therefore, attention should be paid to the management of ecological assets of the ecosystems under rocky desertification control, and the ecological products supply capacity of the ecosystems under rocky desertification control should be improved.

#### 4.2.2. The Priority of Ecological Assets’ Quality Assessment Should Be Higher Than the Value Assessment

The karst rocky desertification area has an important ecological security status, fragile ecological environment and prominent ecological problems. The quantitative assessment of ecological assets is the basis and premise for the rational allocation of ecological resources and adequate protection of the ecological environment [116,117]. The value evaluation of ecological assets is based on the subdivision of ecological assets and the preferred value evaluation method. However, the current value estimation method of ecological assets has defects in its accuracy and spatial expression, and the value evaluation results minimally support regional development. However, the quality and function of their ecological assets can be reflected by evaluation indicators, such as net primary productivity, vegetation coverage and land coverage.

Therefore, we should focus on evaluating the ecological asset quality of the ecosystems under rocky desertification control to support regional development.

#### 4.2.3. The Influencing Mechanism of Ecological Assets of Ecosystems under Rocky Desertification Control, and Its Optimization and Promotion Paths should Be Explored

Due to the spatial heterogeneity of karst areas, the temporal and spatial evolutions of regional ecological assets have more complex characteristics than the other regions. The spatial and temporal changes in ecological assets in some regions are the result of a combined superposition of natural and human factors. Green and sustainable economic development and rural revitalization are necessary for the comprehensive management and ecological restoration of karst rocky desertification areas. In the karst rocky desertification area, there is an apparent correlation between lithology and the development of rocky desertification. Under different lithology conditions, the evolution of the ecosystem is very different due to the influence of the soil formation rate and the thickness of the soil layer. Elevation and slope are the most basic geomorphological indicators, and are some of the main factors affecting surface runoff, soil erosion and land use and have apparent control effects on the distribution of ecosystems. Soil is the foundation of ecosystem formation and evolution, and there is a clear correlation between rocky desertification and soil types in karst areas. Reclamation and cultivation activities in karst rocky desertification areas have long been one of the most critical factors affecting the regional ecology. The cultivated land area can intuitively reflect the impact of human reclamation and cultivation activities on changes in regional ecosystem types and landscape composition. Different rocky desertification degrees and their changes can indicate the long-term ecological system status and intensity of human activities in the region [118]. The rocky desertification control project can be regarded as a positive intervention measure for human activities to improve the quality of the regional ecological environment. Over the years, the rocky desertification control project has played an essential role in promoting the improvement of the regional ecological environment and positive succession of the ecosystem.

Therefore, from the perspective of the coupling of human beings and the land, the natural factors, such as lithology, soil, elevation, slope and landform, which are obviously related to the development of rocky desertification, should be comprehensively considered, in addition to artificial factors, such as farming activities, rocky desertification intensity and rock desertification control projects. The individual factors of the ecological assets’ changes in the rocky desertification control environment should be identified, and the interaction of multiple influencing factors and their contributions to ecological assets’ changes should be decomposed by adjusting the dominant influencing factors of ecological assets to optimize and improve ecological assets.

#### 4.2.4. Strengthen the Research on the Village Ecosystem of Rocky Desertification Control

Due to the strong spatial-scale characteristics of ecosystem services, the results of large- and medium-scale ecosystem studies cannot be directly translated and applied to small scales. The village ecosystem is derived from the concept of “ecosystem”, a natural, economic and socio-ecological network complex formed by human beings, resources and various environmental factors in village areas [110,119]. It consists of the landscapes, such as forests, grasslands, water, farmland, gardens, rural buildings and roads, forming unique landscape, livelihood and consumption structures through mechanisms, such as material circulation, energy flow and information transfer, performing ecological, production and life functions. The village ecosystem is characterized by “health”, “green” and “ecology” [91], with broad, green, open spaces, important air purification, climate regulation, water conservation and other ecological service functions, and is also an indispensable part of ensuring the high-quality and sustainable development of the region [120,121]. However, constrained by their natural environment, villages in karst areas have the characteristics of a fragile ecological environment, lagging economic development, high incidence of poverty and relatively closed regions [122]. It is necessary to strengthen the research conducted on villages in karst rocky desertification control areas. The impact of some ecosystem services extends beyond the limits of administrative boundaries [123]. A single village usually cannot constitute an independent ecological unit in terms of the ecological environment. From the perspective of an ecological environment unit, taking the watershed as the smallest basic regional unit can more scientifically and reasonably reflect the ecological environment of a region. 

Therefore, in karst rocky desertification control areas, we should delineate the boundaries of the village ecosystem according to the watershed, use the delineated range of the village ecosystem as the basic research unit and strengthen the research on the village ecosystem.

## 5. Conclusions

In the current paper, we conducted a systematic literature review by analyzing 117 articles retrieved from the Web of Science, CNKI and Foreign Journal Resource Service System of Guizhou Normal University Library. The main conclusions are as follows: (1) the research conducted on the ecological assets and ecological products supply gradually increases, showing broad research prospects, and (2) among the studies conducted on theoretical, evaluation, mechanism and strategy research, the studies concerning evaluation research are the most common, which is the focus of the current research. Based on the analysis of the research status and progress, four key issues to be solved were proposed, which point out the direction for further research in the future. The future directions of the research conducted on ecological assets and ecological products supply can be performed based on the following aspects: (1) the concept of ecological assets, ecological services and ecological products; (2) the evaluation method system of ecological products; (3) the supply mechanism of ecological products from the perspective of endowment and (4) the consist research of municipal and small-scale ecosystems. The future directions of rocky desertification control projects should focus on the following aspects: (1) pay attention to the management of ecological assets and improve the supply capacity of the ecological products of the ecosystems under rocky desertification control; (2) attach importance to the ecological asset’s quality assessment of ecosystems under rocky desertification control; (3) explore the influencing mechanism of ecological assets on ecosystems under rocky desertification control and its optimization and promotion paths and (4) strengthen the research on village ecosystems under rocky desertification control.

## Figures and Tables

**Figure 1 ijerph-19-10168-f001:**
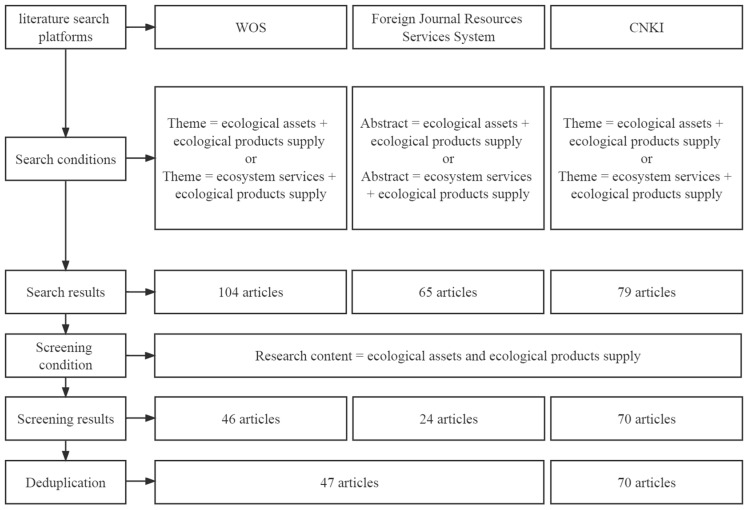
The process of the literature search and screening.

**Figure 2 ijerph-19-10168-f002:**
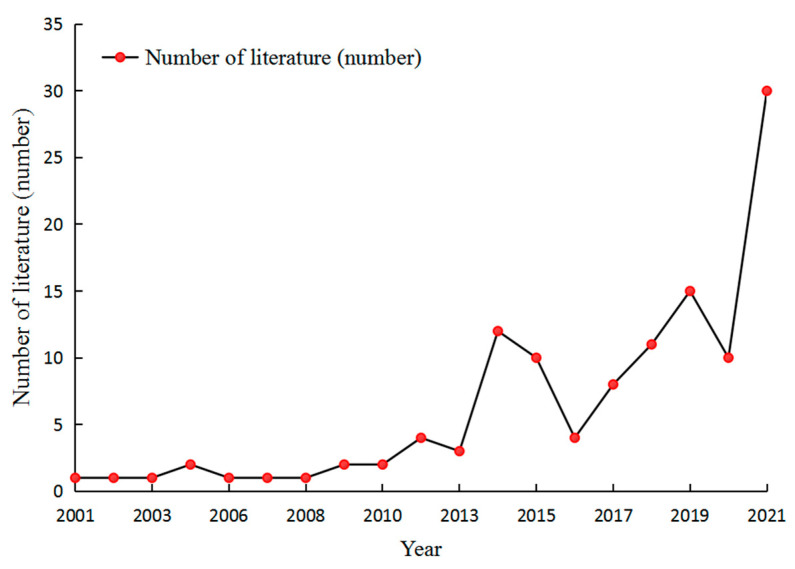
Annual distribution of the literature.

**Figure 3 ijerph-19-10168-f003:**
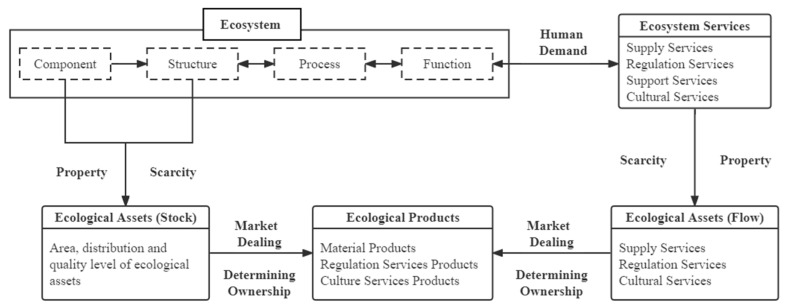
The ecological process of forming ecological assets and products.

**Figure 4 ijerph-19-10168-f004:**
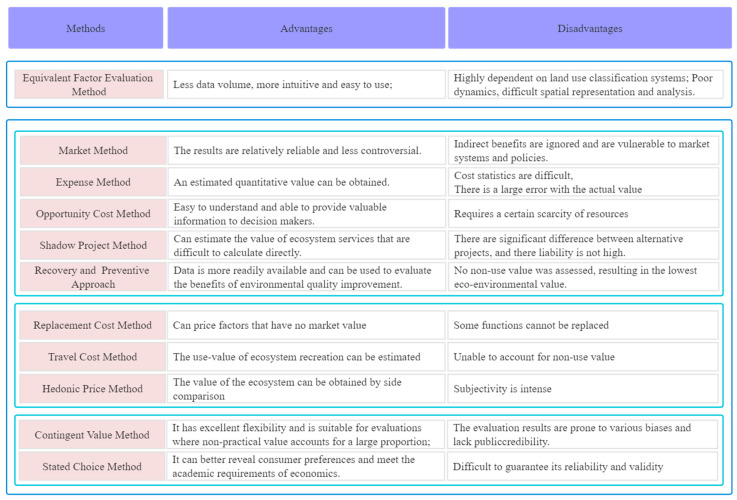
Evaluation methods of ecological product values.

**Table 1 ijerph-19-10168-t001:** The concept developments of ecological assets.

Year	Researcher	Definition
2003	Baiming Chen	Ecological assets are the ecological landscape entities over which the owner takes ownership and obtains economic benefits from, emphasizing the ownership of ecological assets [34].
2004	Kefa Zhou	Ecological assets are the sum of the direct value of biological resources and the function value of ecological services [29].
2004	Yaozhong Pan	Ecological asset is a quantitative value dynamically changing over time, which is the sum of all ecosystem service functions provided by all ecosystem types in the region and the value of their natural resources, and changes to the kind, area and quality of the ecosystem contained in the region [30].
2006	Dan Hu	Adaptive and evolutionary ecological entities formed by the interaction of human or living organisms and their environment serving specific ecosystem economic goals can produce products and services in the future, emphasizing ecological assets’ dynamic structure and function [36].
2007	Jixi Gao	The combination of the value of natural resources and ecosystem services should include all the natural resources and ecological environments that can provide services and welfare for human beings [4].
2015	Peng Hou	The collection of tangible service products and other intangible service products provided by the ecosystem for society is generally measured by the value [31].
2017	Zhiyun OuYang	Ecological assets are the main components of natural resource assets that can provide ecological products and services for human beings, including forests, shrubs, grasslands and wetlands and deserts, and other natural ecosystems, farmland, urban green space and other artificial ecosystems based on natural ecological processes, as well as wildlife resources [35].
2018	Yanxu Liu	Ecological assets are the intersection of the value of natural capital and ecosystem services [32].

**Table 2 ijerph-19-10168-t002:** The concept developments of ecosystem services.

Year	Researcher	Definition
1974	Holdren and Ehrlich	The effects of ecosystems on society and its effectiveness [39].
1997	Costanza	The benefits human populations derive, directly or indirectly, from ecosystem functions [40].
1997	Daily	The conditions and processes through which natural ecosystems, and the species that make them up, sustain and fulfill human lives [41].
2002	De Groot	Products and services that meet human needs are provided either directly or indirectly through the capabilities of natural processes and their components [42].
2005	MA	Various benefits that humans receive from ecosystems include provision services, support services, regulation services and cultural areas [38].
2007	Boyd	Ecosystem services are not the benefits humans obtain from ecosystems, but rather the ecological components directly consumed or enjoyed to ameliorate wellbeing [43].
2008	Wallace	Ecosystem services only include various ecological resources that humans can directly utilize, such as food, wood, drinking water and cultural values. Ecological processes are not ecosystem services [44].
2009	Fisher	The benefits that nature brings to families, communities and the economy [45].
2011	CICES	The contribution of ecosystems to wellbeing [46].
2012	Burkhard	Contribution to ecosystem structure and function: combined with other inputs of human welfare [47].

**Table 3 ijerph-19-10168-t003:** The concept developments of ecological products.

Year	Researcher	Definition
2001	Zijie Fang	Ecological products are produced following the principle of sustainable development, according to a specific production model, and approved by particular agencies, as safe, high-quality and nutritious products; ecological products are characterized by advocating nature, practical, effective, economical and moderate aspects [54].
2008	Jiuxing Zhu	A series of tangible and intangible objects formed through human, conscious, behavioral activities that change (or improve) the whole or pattern of the organism and its relationship with the environment [52].
2010	The National Main Function Zone Planning	Ecological products refer to the natural elements that maintain ecological security, guarantee ecological regulation function and provide a suitable living environment, including fresh air, clean water sources and a pleasant climate [50].
2012	Jida Fan	Broad, ecological products include three types, namely, ecological products with natural, material and institutional attributes, which are greatly affected by the environment; natural ecological products refer to the soil, air, water and forest; material ecological products refer to pollution waste disposal and garbage recycling facilities related to environmental treatment; and institutional ecological products refer to systems and regulations concerning ecological environment protection [55].
2014	Xiangang Zeng	Ecological products refer to the natural elements that maintain life support systems, ensure ecological regulation function and provide environmental comfort, including clean air, clean water, pollution-free soil, lush forests and a suitable climate [48].
2014	Ci Chen	Based on identifying the definition of ecological products proposed by the national main functional area planning research group, it is believed that such products are divided into two categories: tangible (such as organic food, ecological industrial and agricultural products, etc.) and intangible (such as beautiful environment, pleasant climate, environmental safety, etc.) environmental products [49].
2015	Jin Zhang	Ecological products come from natural ecosystems. With this system, human beings obtain relevant products and services of a public nature, and all these obtained products constitute the connotation and scope of ecological products [51].
2019	Linbo Zhang	The final product or service provided for wellbeing through biological production and working together with human production is a daily necessity simultaneous to agricultural and industrial products to meet the needs of humanity for a better life [53].
2021	Linbo Zhang	Ecosystem biological production and human social production provide terminal products or services for the use and consumption of society, including ensuring a good living environment, maintaining ecological security, providing material raw materials and spiritual and cultural services [56].

## Data Availability

Not applicable.

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
