# Peer review of "A Review of Ecological Assets and Ecological Products Supply: Implications for the Karst Rocky Desertification Control"

_ijerph, 2022, doi:10.3390/ijerph191610168_

Round 1

Reviewer 1 Report

Researching ecological assets and the supply of ecological products contributes to the sustainable development of natural-social-economic complex systems. In view of insufficient researches and summary for ecological assets and ecological products supply, this paper quantitatively analyzed and summarized the main progress and results of ecological assets and ecological products supply, and proposed four key scientific issues to be solved and four implications for the karst rocky desertification control. In general, this is a very interesting and meaningful job for future studies. There are still some points worthy of further explanation, which need to be explained in the manuscript.

1. Section materials and methods. the time range of literature retrieval should be clearly stated.

2. Please confirm that the number of results of the literature search is accurate. For example, in CNKI and WOS, according to the retrieval methods and keywords provided by the authors, the initial number of literature obtained is tens of thousands, not just only 104, 65, or 79. It is a very strange result. Objectively speaking, in recent decades, there are a lot of international studies on ecological assets and/or ecological products supply, even in China, there are also a lot of results reported. Thus, maybe, Please explain the reason why only dozens of articles were retrieved.

3. Line 133-134: the third stage, the trend of rapid growth was obtained only based on the data of the number of articles published in 2 years, which was not appropriate.

4. Line 147-157: Does the author just focus on relevant research in China? Relevant international studies are not considered? What about the distribution of foreign scholars and institutions? As far as I know, many scholars and institutions around the world are also carrying out such researches.

5. Line 179 (Table 1): Many international scholars have also defined the concept of ecological assets, so why not adopt the reference in this paper? Does this paper only focus on relevant research in China?

6. Section 4.1.3 Mechanism research. a review of mechanism researches should not only explain what impact factors affect ecosystem services, but also more explain "how" and/or "why" these factors affect ecosystem services. Please strengthen the in-depth analysis of this section.

7. Section “4.3. Implications for the Karst rocky Desertification Control”. in regard to implications for future research, it should not just talk in generalities the mutual problems of insufficient researches on ecological assets and ecological products supply. It is suggested that the authors combine the characteristics of karst areas and karst rocky desertification deeply, and put forward more specific and targeted "unique implications".

8. The citation of references are not standard, for example, Line 253, Line 266, Line 267, et cl.

Author Response

Response to Reviewer 1 Comments

We are grateful and honored to receive your review comments. First of all, I would like to express my sincere gratitude for your hard work on the author's paper. Your constructive review comments have greatly enriched the author's research paper and provided the author with many new ideas and professional help, which is the best affirmation of the author's research work; I fully accept your review comments, according to your relevant suggestions, the author has made significant adjustments and changes in the corresponding parts of the newly submitted paper, as detailed below.

Point 1: Section “materials and methods”. the time range of literature retrieval should be clearly stated.

  • Response 1: Thank you for your sincere advice. We thank the reviewer for pointing out this issue. We added the time range of literature retrieval. The detailed modifications are as follows: The time range for our retrieval is the maximum time range, as at 31 March 2022.

Point 2: Please confirm that the number of results of the literature search is accurate. For example, in CNKI and WOS, according to the retrieval methods and keywords provided by the authors, the initial number of literature obtained is tens of thousands, not just only 104, 65, or 79. It is a very strange result. Objectively speaking, in recent decades, there are a lot of international studies on ecological assets and/or ecological products supply, even in China, there are also a lot of results reported. Thus, maybe, Please explain the reason why only dozens of articles were retrieved.

Response 2: Thank you for your sincere advice. We thank the reviewer for pointing out this issue. At present, there are indeed many literatures that only study ecological assets or ecosystem services, but there are not many studies that combine them with the supply of ecological products. The results of the literature search are correct. If we only search for documents that study ecological assets or ecosystem services, there are indeed tens of thousands of articles. However, if we use ecological products supply to search again on this basis, the obtained documents will not only study ecological assets but also research the supply of ecological products, therefore, the number of literatures found is small.

Point 3: Line 133-134: “the third stage”, the trend of rapid growth was obtained only based on the data of the number of articles published in 2 years, which was not appropriate.

Response 3: Thank you for your sincere advice. We thank the reviewer for pointing out this issue. We have corrected these problems. The detailed modifications are as follows:

…Figure 2 shows that the period between 2000 and 2021 can be roughly divided into three stages. At the first stage (2000–2010), the number of documents is generally low, and the annual number of documents was not more than 5, as starting stage. At the second stage (2011–2021), the number of documents increased, at the stage of fluctuating growth stage.

Point 4: Line 147-157: Does the author just focus on relevant research in China? Relevant international studies are not considered? What about the distribution of foreign scholars and institutions? As far as I know, many scholars and institutions around the world are also carrying out such researches.

Response 4: Thank you for your sincere advice. We thank the reviewer for pointing out this issue. For this problem, we will explain it as follows.

The concept of ecological products has been put forward in China by the Chinese, and some pilot projects in China have formed models that are worth learning from; (2) international scholars use the concept of ecosystem services more than ecological products. The eco-products related academic research is still in the exploratory stage. International institutions and scholars have carried out much research, but they are relatively scattered, and the number of papers published by each institution is less than two. Among all research institutions, there are more international research institutions than Chinese research institutions, but most of the research institutions ranked high in the number of documents are Chinese institutions. Only the top 10 institutions and scholars by the number of publications are listed here, so there are no international institutions and scholars.

  1. Line 179 (Table 1): Many international scholars have also defined the concept of ecological assets, so why not adopt the reference in this paper? Does this paper only focus on relevant research in China?

Response 5: Thank you for your sincere advice. We thank the reviewer for pointing out this issue. The concept of ecological assets has been put forward in China by the Chinese; International scholars use the concept of ecosystem services more than ecological assets. Therefore, the definition of ecological assets by international scholars is not reflected in Table 1, and international scholars use the concept of ecosystem services more. Table 2 shows the definitions of ecosystem services by international scholars.

  1. Section “4.1.3 Mechanism research”. a review of mechanism researches should not only explain “what impact factors” affect ecosystem services, but also more explain "how" and/or "why" these factors affect ecosystem services. Please strengthen the in-depth analysis of this section.

Response 6: Thank you for your sincere advice. We thank the reviewer for pointing out this issue. We have corrected these problems. The detailed modifications are as follows (The yellow part is the newly added content):

The research on the influencing factors of ecosystem services can help to simulate future ecosystem service scenarios accurately and then serve the scientific formulation of strategies for improving the supply capacity of ecological products. To this end, many scholars have explored natural factors (climate change, soil, topography, vegetation) [62,66,80] , socioeconomic factors (population, GDP, income, education, policy) [81,82], human activities (land use, urbanization, ecological construction) [83-97] on ecosystem services. Natural factors are the basis of determining the spatial and temporal distribution of ecosystem services. Land use change can affect ecosystem service changes. The uneven regional distribution and diversification of social and economic factors lead to a human preference for different types of ecosystem services, leading to the difference in ecosystem services demand[98]. Since changes in ecosystem services are often driven by the synergy of natural factors and human activities, regression models and correlation analyses are often used to separate their impacts on ecosystem services [99].

  • Impact of natural factors on ecosystem services

Natural factors such as climate, soil, topography, and biological factors are the essential components of ecosystems and geographical units. Ecosystem structure and geospatial patterns affect ecological processes, determining the spatiotemporal distribution of ecosystem services [98]. Climate change affects the provision of ecosystem services through direct or indirect effects. On the one hand, climate change affects the supply level of ecosystem services such as water supply, carbon sink, food supply, net primary productivity, vegetation coverage, and biological distribution by directly changing precipitation and temperature [100-102]. On the other hand, climate change affects the supply level of ecosystem services by changing land use/cover, ecosystem structure and composition, and resource use intensity [93]. Topographic factors control the distribution of spatial hydrothermal resources, which affect many environmental conditions and ecological processes such as actual solar radiation, temperature, soil mineralization rate, and vegetation distribution, and directly determine the supply and maintenance of ecosystem services[98,103]. Li believed that factors such as increased precipitation, increased distance from water sources, and potash fertilizers all had positive effects on improving HulunBuir grassland ecosystem services [62]. Han shows that precipitation, evaporation, and root depth are the main factors affecting the spatial pattern of water supply in Guizhou Province. The spatial heterogeneity of soil consolidation is caused by rainfall erosivity, topography, cover management factors, and support practices of land use types [66].

  • Impact of socioeconomic factors on ecosystem services

The uneven regional distribution and diversified development of social and economic factors lead to a human choice preference for different types of ecosystem services, which in turn leads to the trade-off and demand difference of ecosystem services[98]. The economic development model affects the accumulation and utilization of ecosystem service value by affecting land use, which in turn affects the per capita economic income and the welfare obtained from ecosystem services. Optimizing the mode of economic development and reducing the interference and impact of human activities can enhance the service value of regional ecosystems and promote the coordinated development of the ecological environment and social economy [104]. Zhang believed that the supply of multi-ecosystem services was negatively correlated with population density, employed population, and real GDP per capita, and had a significant positive correlation with total fiscal revenue and the number of teachers [105]. Han believed there is a weak negative correlation between urbanization rate, per capita GDP, population density, and ecosystem services [66].

  • Impact of human activities on ecosystem services

The influence of human activity on ecosystem services is highly complex. One type of human activity can affect multiple ecosystem services; many kinds of human activities can cause affect one ecosystem service [106]. Human activities, such as land use, urbanization, and ecological conservation projects, can change landscape patterns and ecological processes[79]. Human activities affect the ability of ecosystems to maintain biodiversity and provide ecosystem products by changing the earth's habitat, ecosystem structure, and biogeochemical cycles [106]. Land use plays a decisive role in maintaining the ecological environment system. Land use change directly affects the composition and morphology of the ecosystem and ultimately affects the ability of the ecosystem to provide ecosystem services [84]. The urban expansion changes the structure of many ecosystems by changing land use patterns [86,87]. Zhang believed that urban expansion transforms natural and semi-natural land cover into urban land (cultivated land into the artificial surface), which has a powerful negative impact on the ecosystem service value of urban agglomerations along the coast of the Bohai Rim, China [101]. Ecological restoration projects can lead to significant changes in land use and land cover changes and have been recognized by the international community as an essential mean to enhance and maintain biodiversity and ecosystem services [107]. Gao and Zhang believe that karst rocky desertification governance ecological restoration projects have improved ecosystem services [89,90].

  1. Section “4.3. Implications for the Karst rocky Desertification Control”. in regard to implications for future research, it should not just talk in generalities the mutual problems of insufficient researches on ecological assets and ecological products supply. It is suggested that the authors combine the characteristics of karst areas and karst rocky desertification deeply, and put forward more specific and targeted "unique implications".

Response 7: Thank you for your sincere advice. We thank the reviewer for pointing out this issue. We have rewritten this section. The detailed modifications are as follows:

4.2. Implications for the karst rocky desertification control  

The problem of karst rocky desertification lies in the loss of surface soil, exposed bedrock, loss of land productivity, and degradation of the ecological environment caused by soil erosion [121]. The fundamental cause of karst rocky desertification is that humans ignore the ecosystem's intangible services, care more about the specific products provided by the ecosystem and carry out non-renewable plunder. Rocky desertification control projects should increase the area of the ecosystem, improve the quality of the ecosystem, and improve the ability of the ecosystem to provide ecological products and services. Based on the above analysis, there are mainly the following implications for the rocky desertification control.

4.2.1. Focus on ecological assets management to promote the supply capacity of ecological products

At the end of the 20th century, the Chinese government launched a national comprehensive rocky desertification control program [25], forming rocky desertification control ecosystems in which man and nature interact. The rocky desertification control ecosystem under takes critical ecological functions such as water conservation, soil and water conservation, and biodiversity maintenance in karst areas. However, the rocky desertification control ecosystem has a simple structure, poor stability and weak resistance, and the control results are difficult to maintain [122-124], which profoundly affects the supply capacity of ecosystem services and products. Regarding the definition of ecological assets, the author agrees with Ouyang Zhiyun's point of view, that ecological assets are all kinds of ecosystems and wildlife resources.Therefore, in rocky desertification control areas, the rocky desertification control ecosystem can be regarded as ecological assets. The ecosystem produces supply, regulation, support and cultural services through ecological processes. Ecological products should include the total of ecological material products, ecological regulation services and ecological cultural services obtained by human beings from the rocky desertification control ecosystem. Taking the rocky desertification control forest ecosystem as an example, the ecological products obtained by human beings include: prickly pears, plums, walnuts, loquats, prickly ash and other ecological material products; ecological regulation services such as water conservation, carbon fixation and oxygen release, and water purification; ecological cultural services such as leisure and tourism.

Therefore, attention should be paid to the management of ecological assets of the rocky desertification control ecosystem, and the ecological products supply capacity of the rocky desertification control ecosystem should be improved.

4.2.2. The priority of ecological assets quality assessment should be higher than value assessment

The ecological security status, fragile ecological environment, and significant ecological problems in the rocky desertification areas in karst are essential. Quantitative assessment of ecological assets is the basis and premise for rational allocation of ecological resources and adequate protection of the ecological environment [125,126]. The value evaluation of ecological assets is based on the subdivision of ecological assets and the preferred value evaluation method. However, the current value estimation method of ecological assets has defects in accuracy and spatial expression, and the value evaluation results are of a minimal support role for regional development. However, the quality and function of their ecological assets can be reflected by the evaluation indicators, such as net primary productivity, vegetation coverage, land coverage.

Therefore, we should focus on evaluating the ecological asset quality of the rocky desertification control ecosystem to support regional development.

4.2.3. The influencing mechanism of ecological assets of rocky desertification control ecosystem and its optimization and promotion paths should be explored

Due to the spatial heterogeneity of karst areas, regional ecological assets' temporal and spatial evolution has more complex characteristics than other regions. Regional ecological assets' temporal and spatial changes result from the comprehensive superposition of natural and human factors. It is significant for comprehensive management of rocky desertification and ecological restoration, green and sustainable economic development, and rural revitalization in karst rocky desertification areas. In the karst rocky desertification area, there is an apparent correlation between the lithology and the development of rocky desertification. Under different lithology conditions, the evolution of the ecosystem is very different due to the influence of the soil formation rate and the thickness of the soil layer. Elevation and slope are the most basic geomorphological indicators and are one of the main factors affecting surface runoff, soil erosion, and land use; and have apparent control effects on the distribution of ecosystems. Soil is the foundation of ecosystem formation and evolution, and there is a clear correlation between rocky desertification and soil types in karst areas. Reclamation and cultivation activities in karst rocky desertification areas have long been one of the most critical factors affecting the regional ecology. The cultivated land area can intuitively reflect the impact of human reclamation and cultivation activities on changes in regional ecosystem types and landscape composition. Different rocky desertification degrees and their changes can indicate the long-term ecological system status and the intensity of human activities in the region. The rocky desertification control project can be regarded as a positive intervention measure for human activities to improve the quality of the regional ecological environment. Over the years, the rocky desertification control project has played an essential role in promoting the improvement of the regional ecological environment and the positive succession of the ecosystem.

Therefore, from the perspective of coupling between man and land, natural factors such as lithology, soil, elevation, slope and landform that are obviously related to the development of rocky desertification should be comprehensively considered, as well as artificial factors such as farming activities, rocky desertification intensity, and rock desertification control projects. Identify the single factors of the ecological assets changes, decompose the interaction of multiple influencing factors and their contribution to ecological assets change. By adjusting the dominant influencing factors of ecological assets to optimize and improve ecological assets.

4.2.4. Strengthen the research on the village ecosystem of rocky desertification control

The village ecosystem is derived from the concept of "ecosystem", a natural, economic and socio-ecological network complex formed by human beings, resources and various environmental factors in village areas [120,127]. It consists of forest, grassland, water, farmland, garden, rural architecture and roads, and other landscapes, forming a unique landscape structure, livelihood structure and consumption structure through mechanisms such as material circulation, energy flow and information transfer, performing ecological functions, production functions and life functions. The village ecosystem is characterized by "health", "green" and "ecology" [128], with broad green open space, important air purification, climate regulation, water conservation and other ecological service functions, and is also an indispensable part of ensuring the high-quality and sustainable development of the region [129,130]. However, constrained by their natural environment, villages in karst areas have the characteristics of fragile ecological environment, lagging economic development, high incidence of poverty, and relatively closed regions [131]. It is necessary to strengthen the research on villages in karst rocky desertification control areas. The impact of some ecosystem services extends beyond the limits of administrative boundaries [132]. A single village usually cannot constitute an independent ecological unit in terms of ecological environment. From the perspective of ecological environment unit, taking the watershed as the smallest basic regional unit can more scientifically and reasonably reflect the ecological environment of a region. Therefore, we should delineate the boundary of the village ecosystem according to the watershed, and take the delineated range of the village ecosystem as the basic research unit.

Therefore, in karst rocky desertification control areas, we should delineate the boundaries of the village ecosystem according to the watershed, take the delineated range of the village ecosystem as the basic research unit, and strengthen the research on the village ecosystem.

  1. The citation of references are not standard, for example, Line 253, Line 266, Line 267, et cl.

Response 8: Thanks for your comment, we have checked and revised the reference format.

Reviewer 2 Report

Abstract

Most of the abstract is one sentence. It needs to be broken down and converted into a standard grammatical structure. The content is ok, but grammar needs considerable work.

Introduction

Although the principle of a karst landscape is introduced, the specific focus of interest to the writers is not. If a significant part of the discussion is to be about a particular place, then that place and the reasons for discussing it need to be properly introduced. As currently structured, it is not clear if this paper is intended to be a general review of ecosystem services literature (which is most of the content), or a commentary on the services provided by a particular landscape. The second of these is only poorly addressed and that content should probably be removed.

Methods

Straightforward, apart from some difficult English. The Methods used to assign a paper to a particular category are not described. It is unlikely that all the papers fitted easily into one category or another.

Results.

Most of the results are actually in the discussion. The detail provided in the three Results sections are of little interest and most of it can be removed.

Discussion

Much of this content is actually results. The general content is interesting, but the arguments are sometimes very difficult to follow. Some sections cannot be understood at all. The Discussion needs to be reconstructed to separate results and discussion comments. Or, the paper should be written in more of an essay style.

Some very long and complex sentences need editing.

All references to “abroad” should be removed for content of an international journal. Presumably this means “outside China”, but that is never defined. Nor is it appropriate. In line 192 there is reference to a State Council and in line 193 reference to the “eighteenth big connotation”. The reader has no understanding of these references or why they are included.

There are strong human-centric themes running through the paper. Just one of the definitions in Table 2 includes the possibility that nature provides services to itself rather than specifically to humans. While there is no question that the notion of ecosystem services has been developed by humans in order to explore the economic and other values that nature provides, there is a broader perspective that should not be lost. Nature does not exist in order to service humans, despite the messaging in certain religions and the narrow perspective of many political systems. The perspective of aboriginal groups that tend to live “close to nature” is also relevant. There are strong cultural values and themes that run through their relationship with nature that go far beyond economic values or environmental support for human activities. These concepts need stronger messaging in a review of this sort.

The conclusion is written as a summary, and is not a conclusion.

General

Figure 3 is unnecessary as all content is in the text

Considerable attention to grammatical construction and writing clarity is needed. The paper appears to have been translated and an English writer is needed to convert the current writing for clarity.

Author Response

Response to Reviewer 2 Comments

We are grateful and honored to receive your review comments. First of all, I would like to express my sincere gratitude for your hard work on the author's paper. Your constructive review comments have greatly enriched the author's research paper and provided the author with many new ideas and professional help, which is the best affirmation of the author's research work; I fully accept your review comments, according to your relevant suggestions, the author has made significant adjustments and changes in the corresponding parts of the newly submitted paper, as detailed below.

Point 1: Abstract:

Most of the abstract is one sentence. It needs to be broken down and converted into a standard grammatical structure. The content is ok, but grammar needs considerable work.

Response 1: Thank you for your sincere advice. We thank the reviewer for pointing out this issue. We have corrected these problems. The detailed modifications are as follows:(1) We added the concept of ecological assets; (2) We rewritten the third, fourth, and fifth sentences, and the results (3) and (4). The detailed modifications are as follows:

Ecological assets refer to natural resource assets that can provide ecological products and services for human beings and are also the basis of high-quality ecological products. Researching ecological assets and the supply of ecological products contributes to the sustainable development of natural-social-economic complex systems. This study conducted a literature search and statistical analysis based on Web of Science, CNKI and Foreign Journal Resource Service System of Guizhou Normal University Library literature databases. We reviewed 117 publications on studies of ecological assets and ecological products suuply. Based on summarizing the landmark results, the key scientific issues that need to be solved are proposed, and their implications for karst rocky desertification control are discussed. The results show that: (1) The number of literature published each year from 2001 to 2020 shows a fluctuating growth trend. (2) Research contents concentrate on four dimensions: theoretical research, evaluation research, mechanism research, and strategies research. Among them, evaluation research is the focus and hotspot of the research. (3) It is necessary to deepen the research on the definition and distinction of ecological assets, ecological services, and ecological products; the evaluation system of ecological products; the supply mechanism of ecological products from the perspective of resource endowment; and the research of ecosystem coherence at different scales. (4) The karst rocky desertification control should focus on ecological assets management to promote the supply capacity of ecological products,pay attention to the quality assessment of ecological assets,explore the influencing mechanism of ecological assets and its optimization and promotion paths, and strengthen the research of karst rocky desertification control village ecosystem.

Point 2: Introduction:

Although the principle of a karst landscape is introduced, the specific focus of interest to the writers is not. If a significant part of the discussion is to be about a particular place, then that place and the reasons for discussing it need to be properly introduced. As currently structured, it is not clear if this paper is intended to be a general review of ecosystem services literature (which is most of the content), or a commentary on the services provided by a particular landscape. The second of these is only poorly addressed and that content should probably be removed.

Response 2: Thank you for your sincere advice. The author has carefully considered your comments. For the Introduction section, the author has made the following changes: We have rewritten the third paragraph of the introduction to this paper. We introduced the causes and consequences of rocky desertification, point out the current difficulties in rocky desertification control, and explain the reasons why research on ecological assets and ecological products supply can guide rocky desertification control. The detailed modifications are as follows:

   Ecological environment problems in ecologically fragile areas have always been paid attention to. Karst refers to the dissolution of water on soluble rocks (carbonate rocks, Etc.) and the resulting various landscapes and phenomena on the surface and underground [8,9], which is an apparent binary landform morphological structure [10]. Karst landscapes are widely distributed worldwide, accounting for 15% of the world's total land area [11-13]. Concentrated karsts are mainly distributed in central and southern Europe, eastern North America, and southwestern China [14-15]. The southern China karst area is the most typical, complex landscape type area in the world[16], called the world's karst museum[17]. The karst area is restricted by the unique geological conditions of soluble carbonate rocks, the soil formation is prolonged, the soil layer is thin and discontinuous, and the hydrological process responds rapidly [18]. Unscientific human activities lead to soil erosion in the karst environment, exposing the bedrock under the soil and forming a landscape similar to the desert on the surface [19-23]. This process is called karst rocky desertification [19-23]. Karst rocky desertification has caused problems such as the decline of land productivity, the sharp reduction of surface vegetation coverage, the weakening of system water conservation capacity, the depletion of surface water sources, the loss of land resources, and the reduction of food production [24], resulting in the gradual loss of ecosystem services and threatening the sustainability of the region development. To this end, the Chinese government launched a national comprehensive rock desertification control program at the end of the 20th century[25]. Since then, the expansion of rock desertification, the destruction of vegetation, and soil degradation have been initially alleviated, and phased results have been achieved [26-27]. However, there are still problems such as difficulty in effectively maintaining the governance results, the inability to eliminate the contradiction between man and land, and increasing difficulty in governance when dealing with rocky desertification in China [28]. How to scientifically promote the governance of karst rocky desertification, improve the quality and sustainability of ecological restoration, and enhance the ability of ecosystems to provide ecological products and services have become critical issues for the harmonious coexistence of humans and nature in karst rocky desertification areas. Ecological assets are part of the ecosystem. The management of ecological assets is an effective way to change the structure of the ecosystem, the key to changing the quality of the ecosystem, and an inevitable way to improve the supply capacity of ecological products. Therefore, it is necessary to review the research progress of ecological assets and ecological product supply in order to guide the work of karst rocky desertification control.

Point 3: Methods:

Straightforward, apart from some difficult English. The Methods used to assign a paper to a particular category are not described. It is unlikely that all the papers fitted easily into one category or another.

Response 3: Thank you for your sincere advice. For the Methods section, the author has added methods for assigning a paper to a particular categories. The detailed modifications are as follows:

We divide the research content according to the subject content of the article. If the article only differentiates and analyzes related concepts, it is classified as theoretical research. If the whole article only includes accounting (ecological asset accounting or ecosystem service value or ecological product value accounting or ecological product supply capacity assessment), it will be classified as an evaluation research; if the article analyzes ecological assets or ecosystem service impact factors , it is classified as mechanism research; if the article conducts research on strategies such as ecological compensation and ecological function zoning on the basis of accounting, it is classified as strategic research.

Point 4: Results:

Most of the results are actually in the discussion. The detail provided in the three Results sections are of little interest and most of it can be removed.

Point 5: Discussion

Much of this content is actually results. The general content is interesting, but the arguments are sometimes very difficult to follow. Some sections cannot be understood at all. The Discussion needs to be reconstructed to separate results and discussion comments. Or, the paper should be written in more of an essay style.

Response 4-5: Thank you for your sincere advice. We have reconstructed the Result and Discussion. The detailed modifications are as follows:

3 Result

3.1 Literature Statistical Analysis

3.2 Main Progress and Landmark Achievements

4 Discussion

4.1 Key scientific Issues to be Solved

4.2 Implications for the Karst rocky Desertification Control  

Point 6: Some very long and complex sentences need editing.

Response 6: Thank you for your comment. We thank the reviewer for pointing out this issue. We have corrected these problems.

Point 7: All references to “abroad” should be removed for content of an international journal. Presumably this means “outside China”, but that is never defined. Nor is it appropriate. In line 192 there is reference to a State Council and in line 193 reference to the “eighteenth big connotation”. The reader has no understanding of these references or why they are included.

Response 7: Thank you for your comment. We thank the reviewer for pointing out this issue. We have corrected these problems. All this content has been removed.

Point 8: There are strong human-centric themes running through the paper. Just one of the definitions in Table 2 includes the possibility that nature provides services to itself rather than specifically to humans. While there is no question that the notion of ecosystem services has been developed by humans in order to explore the economic and other values that nature provides, there is a broader perspective that should not be lost. Nature does not exist in order to service humans, despite the messaging in certain religions and the narrow perspective of many political systems. The perspective of aboriginal groups that tend to live “close to nature” is also relevant. There are strong cultural values and themes that run through their relationship with nature that go far beyond economic values or environmental support for human activities. These concepts need stronger messaging in a review of this sort.

Response 8: Thank you for your comment. We thank the reviewer for pointing out this issue.

We fully agree with your point of view. Cultural services are an important part of ecological assets, ecosystem services, and ecological products. Therefore, we supplement the content of cultural services in the Table 2 and section 4.2.1. The detailed modifications are as follows:

Various benefits that humans receive from ecosystems, include provision services, support services, regulation services and cultural areas [42].

Therefore, in rocky desertification control areas, the rocky desertification control ecosystem can be regarded as ecological assets. The ecosystem produces supply, regulation, support and cultural services through ecological processes. Ecological products should include the total of ecological material products, ecological regulation services and ecological cultural services obtained by human beings from the rocky desertification control ecosystem. Taking the rocky desertification control forest ecosystem as an example, the ecological products obtained by human beings include: prickly pears, plums, walnuts, loquats, prickly ash and other ecological material products; ecological regulation services such as water conservation, carbon fixation and oxygen release, and water purification; ecological cultural services such as leisure and tourism.

Point 9: The conclusion is written as a summary, and is not a conclusion.

Response 9: Thank you for your sincere advice. We have rewritten the third paragraph of the conclusion. The detailed modifications are as follows:

In this paper, we conducted a systematic literature review by analyzing 117 articles retrieved from the Web of Science, CNKI and Foreign Journal Resource Service System of Guizhou Normal University Library. The main conclusions are as follows: (1) The research on ecological assets and ecological products supply is gradually increasing, showing broad research prospects; (2) Among the studies on theoretical research, evalu-ation research, mechanism research, and strategy research, the studies on evaluation re-search are most common, which is the research focus. Based on the analysis of the re-search status and progress, four key issues to be solved are put forward, which point out the direction for further research in the future. The future directions of ecological assets and ecological products supply can be conducted based on the following aspects: (1)the concept of ecological assets, ecological services, and ecological products; (2)evaluation method system of ecological products; (3)the supply mechanism of ecological products from the perspective of endowment; (4)the consistency research of municipal and small-scale ecosystems. The future directions of rocky desertification control projects should focus on the following aspects: (1) Pay attention to the management of ecological assets and improve the supply capacity of the ecological products of the rocky desertification control ecosystem; (2) Attach importance to the ecological assets quality assessment of rocky desertification control ecosystem; (3) Explore the influencing mechanism of ecological assets of rocky desertification control ecosystem and its optimization and promotion paths; (4) Strengthen the research on the rocky desertification control village ecosystems.

Point 10: Figure 3 is unnecessary as all content is in the text.

Response 10: Thank you for your comment. We thank the reviewer for pointing out this issue. We have removed Figure 3.

Point 11: Considerable attention to grammatical construction and writing clarity is needed. The paper appears to have been translated and an English writer is needed to convert the current writing for clarity.

Response 11: Thank you for your comment. We thank the reviewer for pointing out this issue. For the language problem of the article, the author analyzes the full text sentence by sentence with the help of our peers with strong English expression skills, and corrects the errors.

Reviewer 3 Report

The paper is designed as a review of publications about the assessment of ecosystem assets, products and services, which is nowadays an important issue. However, the topic is not analyzed well.

·       The abstract is rather unclear. (1) A short (one sentence) definition of ecological assets would be suitable; (2) it is unclear from the abstract if it is a review of publications about ecological assets, products and services in general or it is a case study of the karst rocky desertification control.

·       The part Methods can be shortened. Really the Fig.1 explains well the way of searching literature.

·       In the Introduction it would be good (1) enumerate some examples of ecological assets and (2) to describe briefly, what do you mean under “rocky desertification” and its control. Maybe “control” is not a good term, as desertification is an unwanted process and, as I can assume, it is not about controlled desertification, but about it prevention or mitigation.

·       In the chapter 4.1.1, Tables 1-3 gave a lot of definitions of ecological assets, products and services, which sound rather similar, without almost any analysis (maybe one sentence in lines 222-25). In the Discussion the authors should compare these definitions and concepts, notice what is common or different and give clear explanations, how these concepts are displayed in the real life. E.g., if I pick up mushrooms in the forest – it is ecosystem product (I get food), ecosystem service (I get pleasure and fresh air) and ecosystem asset (only if I do it in my own private forest) – did I understood you well?

·       Also the literature review included international databases, foreign literature is rather poorly presented in the Discussion.

·       Fig. 5 is one of the most important things in the paper. It is generally good, but maybe it would be good to add one sentence of explanation to each method, because it is not always clear from its name, in what it consists.

·       The whole part 4 about rocky desertification areas contains nothing specific for this type of landscape. It can be easily omitted.

·       The part 4.3.4 about village ecosystems is rather unfounded. What particularly and why must be studied at such a small scale? Finally, the whole story is aimed to provide information for legislators and policy makers, and regulations could not differ from village to village.

P   Please see attached the comments within the manuscript. 

Author Response

Response to Reviewer 3 Comments

We are grateful and honored to receive your review comments. First of all, I would like to express my sincere gratitude for your hard work on the author's paper. Your constructive review comments have greatly enriched the author's research paper and provided the author with many new ideas and professional help, which is the best affirmation of the author's research work; I fully accept your review comments, according to your relevant suggestions, the author has made significant adjustments and changes in the corresponding parts of the newly submitted paper, as detailed below.

Point 1: The abstract is rather unclear. (1) A short (one sentence) definition of ecological assets would be suitable; (2) it is unclear from the abstract if it is a review of publications about ecological assets, products and services in general or it is a case study of the karst rocky desertification control.

Response 1: Thank you for your sincere advice. We thank the reviewer for pointing out this issue. This paper is a review of publications about ecological assets, products and services in general. Through a systematic review of 117 publications, some insights into rocky desertification control are put forward. For the abstract sections, the author has made the following changes:(1) We added the concept of ecological assets based on your suggestion.  (2)We rewrite the third, fourth, and fifth sentences, and the results (3) and (4). The detailed modifications are as follows (The yellow part is the modified content):

Ecological assets refer to natural resource assets that can provide ecological products and services for human beings and are also the basis of high-quality ecological products. Researching ecological assets and the supply of ecological products contributes to the sustainable development of natural-social-economic complex systems. This study conducted a literature search and statistical analysis based on Web of Science, CNKI and Foreign Journal Resource Service System of Guizhou Normal University Library literature databases. We reviewed 117 publications on studies of ecological assets and ecological products suuply. Based on summarizing the landmark results, the key scientific issues that need to be solved are proposed, and their implications for karst rocky desertification control are discussed. The results show that: (1) The number of literature published each year from 2001 to 2020 shows a fluctuating growth trend. (2) Research contents concentrate on four dimensions: theoretical research, evaluation research, mechanism research, and strategies research. Among them, evaluation research is the focus and hotspot of the research. (3) It is necessary to deepen the research on the definition and distinction of ecological assets, ecological services, and ecological products; the evaluation system of ecological products; the supply mechanism of ecological products from the perspective of resource endowment; and the research of ecosystem coherence at different scales. (4) The karst rocky desertification control should focus on ecological assets management to promote the supply capacity of ecological products,pay attention to the quality assessment of ecological assets,explore the influencing mechanism of ecological assets and its optimization and promotion paths, and strengthen the research of karst rocky desertification control village ecosystem.

  • Point 2: The part Methods can be shortened. Really the Fig.1 explains well the way of searching literature.

Response 2: Thank you for your sincere advice. For the Methods sections, the author has made the following changes: We removed the step of document acquisition and added the time range of document search and the method of document classification. The detailed modifications are as follows:

To identify relevant studies, a search was conducted based on the platforms including Web of Science (WOS), Foreign Journal Resource Service System and China National Knowledge Infrastructure (CNKI). The time range for our retrieval is the maximum time range, as at 31 March 2022. Figure 1 showed the process of literature search and screening.We finally obtained 47 English publications and 70 Chinese publications. The top 3 contributors in number of literature of the topic are Huiyi Yu (4), Baiming Chen (3), Xiahui Wang (3). When calculating the number of published literature by authors, all authors in the literature were accounted regardless of the author’s order. The low number of articles published by each author shows the lack of systematic and in-depth research in this field.

We divide the research content according to the subject content of the article. If the article only differentiates and analyzes related concepts, it is classified as theoretical research. If the whole article only includes accounting (ecological asset accounting or eco-system service value or ecological product value accounting or ecological product supply capacity assessment), it will be classified as an evaluation research; if the article analyzes ecological assets or ecosystem service impact factors , it is classified as mechanism research; if the article conducts research on strategies such as ecological compensation and ecological function zoning on the basis of accounting, it is classified as strategic research.

  • Point 3: In the Introduction it would be good (1) enumerate some examples of ecological assets and (2) to describe briefly, what do you mean under “rocky desertification” and its control. Maybe “control” is not a good term, as desertification is an unwanted process and, as I can assume, it is not about controlled desertification, but about it prevention or mitigation.

Response 3: Thank you for your comments. We apologize for the extra work you have added to our inattentive language translation process. The "rocky desertification" is "karst rocky desertification" in the paper. By reviewing the existing publications, we found that scholars mostly use the term rocky desertification control. Karst rocky desertification control is a measure whose purpose is to achieve the prevention or mitigation of karst rocky desertification, so we used the term rocky desertification control. For the Introduction sections, the author has made the following changes: (1) We added the examples of ecological assets based on your suggestion. (2) We described “karst rocky desertification” and its control. The detailed modifications are as follows: 

The ecological assets stock refers to the ecosystem itself, such as forest ecosystem, grassland ecosystem. The ecological assets flow refers to the product of the continuous output of the ecosystem, or the effect that occurs continuously due to its existence, that is, the supply service, the regulation service, and the cultural service in the ecosystem service [5]. For example, forest product supply, water conservation services, leisure and tourism.

Karst refers to the dissolution of water on soluble rocks (carbonate rocks, Etc.) and the resulting various landscapes and phenomena on the surface and underground [8,9], which is an apparent binary landform morphological structure [10]. Karst landscapes are widely distributed worldwide, accounting for 15% of the world's total land area [11-13]. Concentrated karsts are mainly distributed in central and southern Europe, eastern North America, and southwestern China [14-15]. The southern China karst area is the most typical, complex landscape type area in the world[16], called the world's karst museum[17]. The karst area is restricted by the unique geological conditions of soluble carbonate rocks, the soil formation is prolonged, the soil layer is thin and discontinuous, and the hydrological process responds rapidly [18]. Unscientific human activities lead to soil erosion in the karst environment, exposing the bedrock under the soil and forming a landscape similar to the desert on the surface [19-23]. This process is called karst rocky desertification [19-23]. Karst rocky desertification has caused problems such as the decline of land productivity, the sharp reduction of surface vegetation coverage, the weakening of system water conservation capacity, the depletion of surface water sources, the loss of land resources, and the reduction of food production [24], resulting in the gradual loss of ecosystem services and threatening the sustainability of the region development. To this end, the Chinese government launched a national comprehensive rock desertification control program at the end of the 20th century[25]. Since then, the expansion of rock desertification, the destruction of vegetation, and soil degradation have been initially alleviated, and phased results have been achieved [26-27].

  • Point 4:In the chapter 4.1.1, Tables 1-3 gave a lot of definitions of ecological assets, products and services, which sound rather similar, without almost any analysis (maybe one sentence in lines 222-25). In the Discussion the authors should compare these definitions and concepts, notice what is common or different and give clear explanations, how these concepts are displayed in the real life. E.g., if I pick up mushrooms in the forest – it is ecosystem product (I get food), ecosystem service (I get pleasure and fresh air) and ecosystem asset (only if I do it in my own private forest) – did I understood you well?

Response 4: Thank you for your comments. At present, the definitions of ecological assets and ecological products are not uniform. Based on different definitions, the relationship between ecological assets, ecological products, and ecosystem services is different. We propose that clearly defining the concepts of the three and their interrelationships is a key issue to be solved in the future. However, according to the specific research purpose, the relationship between the three can be briefly discussed. Therefore, in section 4.2.1, the author briefly describes the relationship between ecological assets, ecological products, and ecosystem services in rocky desertification control areas. The detailed modifications are as follows:

Regarding the definition of ecological assets, the author agrees with Ouyang Zhiyun's point of view, that ecological assets are all kinds of ecosystems and wildlife resources.Therefore, in rocky desertification control areas, the rocky desertification control ecosystem can be regarded as ecological assets. The ecosystem produces supply, regulation, support and cultural services through ecological processes. Ecological products should include the total of ecological material products, ecological regulation services and ecological cultural services obtained by human beings from the rocky desertification control ecosystem. Taking the rocky desertification control forest ecosystem as an example, the ecological products obtained by human beings include: prickly pears, plums, walnuts, loquats, prickly ash and other ecological material products; ecological regulation services such as water conservation, carbon fixation and oxygen release, and water purification; ecological cultural services such as leisure and tourism.

  • Point 5:Also the literature review included international databases, foreign literature is rather poorly presented in the Discussion.

Response 5: Thank you for your comments. We have added foreign literature based on your suggestion in the Discussion.

  • Point 6:Fig. 5 is one of the most important things in the paper. It is generally good, but maybe it would be good to add one sentence of explanation to each method, because it is not always clear from its name, in what it consists.

Response 6: Thank you for your comments. We have added a simple explanation for each method based on your suggestion. The detailed revisions are as follows:

The market value method refers to a method of valuing ecosystem products and functions with market prices, and is mainly used for the evaluation of material products produced by ecosystems. From the perspective of consumers, the expense method takes the sum of the relevant expenses actually paid by tourists for tourism as the tourism economic value of scenic resources. Opportunity cost method means that the use of a scheme under the condition of scarcity of resources means that other schemes must be abandoned, and the maximum benefit that may be obtained in the abandoned scheme constitutes the opportunity cost of the scheme. Assuming that a project with similar actual effect but not actually carried out is adopted, the shadow project method of replacing the economic loss of the project to be assessed with the construction cost of the project. The recovery and preventive approach that assesses how much people are willing to spend to avoid damage to ecosystem services or, if damaged, to restore them to their original levels. Replacement cost method is that some ecosystem services can be replaced by man-made systems, and the value of these ecosystem services can be assessed by the cost of replacing the service. The travel cost method by assessing the willingness of individuals to pay an equivalent travel expense in order to obtain consumption of a non-market item. The hedonic price method that estimates the value of environmental quality by assessing the price people pay for the enjoyment of a good environment. The contingent value method to discover how people price certain environmental changes by interviewing the survey directly to the relevant population, the respondents expressing their demand for a specific environmental product or service.

  • Point 7: The whole part 4 about rocky desertification areas contains nothing specific for this type of landscape. It can be easily omitted.

Response 7: Thank you for your comment. We thank the reviewer for pointing out this issue. Based on these issues, we have rewritten this part in depth by combining the characteristics of karst areas and karst desertification. The detailed modifications are as follows:

4.2. Implications for the karst rocky desertification control  

The problem of karst rocky desertification lies in the loss of surface soil, exposed bedrock, loss of land productivity, and degradation of the ecological environment caused by soil erosion [121]. The fundamental cause of karst rocky desertification is that humans ignore the ecosystem's intangible services, care more about the specific products provided by the ecosystem and carry out non-renewable plunder. Rocky desertification control projects should increase the area of the ecosystem, improve the quality of the ecosystem, and improve the ability of the ecosystem to provide ecological products and services. Based on the above analysis, there are mainly the following implications for the rocky desertification control.

4.2.1. Focus on ecological assets management to promote the supply capacity of ecological products

At the end of the 20th century, the Chinese government launched a national comprehensive rocky desertification control program [25], forming rocky desertification control ecosystems in which man and nature interact. The rocky desertification control ecosystem under takes critical ecological functions such as water conservation, soil and water conservation, and biodiversity maintenance in karst areas. However, the rocky desertification control ecosystem has a simple structure, poor stability and weak resistance, and the control results are difficult to maintain [122-124], which profoundly affects the supply capacity of ecosystem services and products. Regarding the definition of ecological assets, the author agrees with Ouyang Zhiyun's point of view, that ecological assets are all kinds of ecosystems and wildlife resources. Therefore, in rocky desertification control areas, the rocky desertification control ecosystem can be regarded as ecological assets. The ecosystem produces supply, regulation, support and cultural services through ecological processes. Ecological products should include the total of ecological material products, ecological regulation services and ecological cultural services obtained by human beings from the rocky desertification control ecosystem. Taking the rocky desertification control forest ecosystem as an example, the ecological products obtained by human beings include: prickly pears, plums, walnuts, loquats, prickly ash and other ecological material products; ecological regulation services such as water conservation, carbon fixation and oxygen release, and water purification; ecological cultural services such as leisure and tourism.

Therefore, attention should be paid to the management of ecological assets of the rocky desertification control ecosystem, and the ecological products supply capacity of the rocky desertification control ecosystem should be improved.

4.2.2. The priority of ecological assets quality assessment should be higher than value assessment

The ecological security status, fragile ecological environment, and significant ecological problems in the rocky desertification areas in karst are essential. Quantitative assessment of ecological assets is the basis and premise for rational allocation of ecological resources and adequate protection of the ecological environment [125,126]. The value evaluation of ecological assets is based on the subdivision of ecological assets and the preferred value evaluation method. However, the current value estimation method of ecological assets has defects in accuracy and spatial expression, and the value evaluation results are of a minimal support role for regional development. However, the quality and function of their ecological assets can be reflected by the evaluation indicators, such as net primary productivity, vegetation coverage, land coverage.

Therefore, we should focus on evaluating the ecological asset quality of the rocky desertification control ecosystem to support regional development.

4.2.3. The influencing mechanism of ecological assets of rocky desertification control ecosystem and its optimization and promotion paths should be explored

Due to the spatial heterogeneity of karst areas, regional ecological assets' temporal and spatial evolution has more complex characteristics than other regions. Regional ecological assets' temporal and spatial changes result from the comprehensive superposition of natural and human factors. It is significant for comprehensive management of rocky desertification and ecological restoration, green and sustainable economic development, and rural revitalization in karst rocky desertification areas. In the karst rocky desertification area, there is an apparent correlation between the lithology and the development of rocky desertification. Under different lithology conditions, the evolution of the ecosystem is very different due to the influence of the soil formation rate and the thickness of the soil layer. Elevation and slope are the most basic geomorphological indicators and are one of the main factors affecting surface runoff, soil erosion, and land use; and have apparent control effects on the distribution of ecosystems. Soil is the foundation of ecosystem formation and evolution, and there is a clear correlation between rocky desertification and soil types in karst areas. Reclamation and cultivation activities in karst rocky desertification areas have long been one of the most critical factors affecting the regional ecology. The cultivated land area can intuitively reflect the impact of human reclamation and cultivation activities on changes in regional ecosystem types and landscape composition. Different rocky desertification degrees and their changes can indicate the long-term ecological system status and the intensity of human activities in the region. The rocky desertification control project can be regarded as a positive intervention measure for human activities to improve the quality of the regional ecological environment. Over the years, the rocky desertification control project has played an essential role in promoting the improvement of the regional ecological environment and the positive succession of the ecosystem.

Therefore, from the perspective of coupling between man and land, natural factors such as lithology, soil, elevation, slope and landform that are obviously related to the development of rocky desertification should be comprehensively considered, as well as artificial factors such as farming activities, rocky desertification intensity, and rock desertification control projects. Identify the single factors of the ecological assets changes, decompose the interaction of multiple influencing factors and their contribution to ecological assets change. By adjusting the dominant influencing factors of ecological assets to optimize and improve ecological assets.

4.2.4. Strengthen the research on the village ecosystem of rocky desertification control

The village ecosystem is derived from the concept of "ecosystem", a natural, economic and socio-ecological network complex formed by human beings, resources and various environmental factors in village areas [120,127]. It consists of forest, grassland, water, farmland, garden, rural architecture and roads, and other landscapes, forming a unique landscape structure, livelihood structure and consumption structure through mechanisms such as material circulation, energy flow and information transfer, performing ecological functions, production functions and life functions. The village ecosystem is characterized by "health", "green" and "ecology" [128], with broad green open space, important air purification, climate regulation, water conservation and other ecological service functions, and is also an indispensable part of ensuring the high-quality and sustainable development of the region [129,130]. However, constrained by their natural environment, villages in karst areas have the characteristics of fragile ecological environment, lagging economic development, high incidence of poverty, and relatively closed regions [131]. It is necessary to strengthen the research on villages in karst rocky desertification control areas. The impact of some ecosystem services extends beyond the limits of administrative boundaries [132]. A single village usually cannot constitute an independent ecological unit in terms of ecological environment. From the perspective of ecological environment unit, taking the watershed as the smallest basic regional unit can more scientifically and reasonably reflect the ecological environment of a region. Therefore, we should delineate the boundary of the village ecosystem according to the watershed, and take the delineated range of the village ecosystem as the basic research unit.

Therefore, in karst rocky desertification control areas, we should delineate the boundaries of the village ecosystem according to the watershed, take the delineated range of the village ecosystem as the basic research unit, and strengthen the research on the village ecosystem.

  •    Point 8: The part 4.3.4 about village ecosystems is rather unfounded. What particularly and why must be studied at such a small scale? Finally, the whole story is aimed to provide information for legislators and policy makers, and regulations could not differ from village to

Response 8: Thank you for your comment. We thank the reviewer for pointing out this issue. Due to the strong spatial scale characteristics of ecosystem services, the results of large and medium scale ecosystem studies cannot be directly translated and applied to small scales. Although regulations do not vary from village to village, village ecosystems provide most of the world’s food and natural resources, carbon sequestration, water filtration and wildlife habitat, as well as maintaining supplies of food, energy and freshwater products for rural and urban populations. The research on the village ecosystem is of great significance and is a necessary choice to ensure the sustainable supply of village ecosystem services.

 Point 9: ... the 117 obtained literatures were... "publications" is better

Response 9: Thank you for your comments. We apologize for the extra work you have added to our inattentive language translation process. We replaced literatures with publications.

 Point 10:"ecological products of rocky desertification control" - what does it mean?

Response 10: Thank you for your comments. We apologize for the extra work you have added to our inattentive language translation process. Ecological products of rocky desertification control means ecological products of rocky desertification control ecosystem.

 Point 11: There are only 24 ecosystems? Maybe TYPES of ecosystems? According to which classification?

Response 11: Thank you for your comments. We apologize for the extra work you have added to our inattentive language translation process. This refers to the selection of 24 ecosystems for evaluation . We have corrected these problems. The detailed modifications are as follows:

The Millennium Ecosystem Assessment pointed out that of 24 evaluated ecosystem types, 15 are being damaged;

 Point 12: “...extremely fragile due to the special characteristics of the rocks...” which characteristics ?

Response 12: Thank you for your comments. We apologize for the extra work you have added to our inattentive language translation process. This refers to the characteristic that carbonate rocks are soluble in water.

 Point 13: “...the country...” Do you mean China or any country in general?

Response 13: Thank you for your comments. We apologize for the extra work you have added to our inattentive language translation process. We replaced the country with China.

 Point 14: “...However, after controlling rocky desertification, the ecosystem is characterized by instability and easy rebound,...” Unclear. What increases instability - desertification or its control ?

Response 14: Thank you for your comments. We have rewritten this section. The detailed modifications are as follows:

To this end, the Chinese government launched a national comprehensive rock desertification control program at the end of the 20th century[25]. Since then, the expansion of rock desertification, the destruction of vegetation, and soil degradation have been initially alleviated, and phased results have been achieved [26-27]. However, there are still problems such as difficulty in effectively maintaining the governance results, the inability to eliminate the contradiction between man and land, and increasing difficulty in governance when dealing with rocky desertification in China [28]. How to scientifically promote the governance of karst rocky desertification, improve the quality and sustainability of ecological restoration, and enhance the ability of ecosystems to provide ecological products and services have become critical issues for the harmonious coexistence of humans and nature in karst rocky desertification areas.

 Point 15: ...Foreign Journal Resource Service System... Is it a Chinese or international system?

Response 15: Thank you for your comments. It is a Chinese system,

 Point 16: “...65 literatures were found...” "publications" is better

Response 16: Thank you for your comments. We apologize for the extra work you have added to our inattentive language translation process. We replaced literatures with publications.

 Point 17: “ ... 46, 24 and 70...” after excluding irrelevant literature ?

Response 17: Thank you for your comment. We thank the reviewer for pointing out this issue. We have corrected these problems. The detailed modifications are as follows:

Then, according to the research content of ecological assets and ecological products supply, irrelevant literature were screened out. 46, 24 and 70 literature were obtained from WOS, the foreign journal resource service system and CNKI, respectively.

 Point 18: Page 6, Lines 189-193. Too long and unclear; consider reformulate.

Response 18: Thank you for your comment. We thank the reviewer for pointing out this issue. We have corrected these problems. The detailed modifications are as follows:

Throughout the academic study of ecological products, scholars on the definition of ecological products has not yet formed a unified cognition. According to whether the human labor, can be divided into two situations. In a narrow sense, ecological products are ecological services, including fresh air, clean water and pleasant climate and other natural products supply, regulating services, support services and cultural services, all natural production to human welfare [33,34,39,41].

 Point 19: “... Ecological assets emphasize clear property rights...” Really, only Baiming Chen's definition mentions ownership (Table 1)

Response 19: Thank you for your comment. We explain this issue as follows:

Clear property rights and scarcity are the social attributes of ecological assets. Although other scholars do not explicitly mention property rights when they define ecological assets, scarcity and property rights are still the social attributes of ecological assets. Ecological assets come from ecological resources. Ecological resources are not equal to ecological assets. Ecological resources can be transformed into ecological assets. The transformation of ecological resources into ecological assets requires certain conditions: scarcity, clear property rights, and the ability to generate income. Scarce ecological resources will stimulate people's desire to possess, and lead to changes in the owners of ecological resources rights and interests, thus transforming ecological resources into ecological assets. Therefore, ecological assets have scarcity and clear property rights.

 Point 20: “...such as some supply and regulation services provided by ecosystems...” which, for example?

Response 20: Thank you for your comment. For example, forest product supply, water conservation services, leisure and tourism.

       Point 21: “…the ecological and the number of tourists …” ecological what?

Response 21: Thank you for your comments. We apologize for the extra work you have added to our inattentive language translation process. The detailed revisions are as follows: The material quality of ecological products is used, that is, the output of various products provided by the ecosystem within a certain period of time, the functional amount of ecological regulation services and the number of ecological tourism tourists are used to characterize the supply capacity of ecological products.

 Point 22: “...Ecological products, in a narrow sense, are ecosystem services... “ So, is the nice landscape an ecological product or service? And wild berries?

Response 22: Thank you for your comments. The nice landscape and wild berries are ecological products or services.

 Point 23: “...increased distance from water sources...”why increased distance from water sources can have positive effect?

Response 23: Thank you for your comments. We apologize for the extra work you have added to our inattentive language translation process. We replaced increased with increased.

Point 24: “...Zhang believed that...” "suggests", notices or "shows" is better

Response 24: Thank you for your comments. We apologize for the extra work you have added to our inattentive language translation process. We replaced suggests with shows.

 Point 25: “Through the assessment and evaluation of ecological assets to understand the quality of ecosystems and their changes, combined with the spatial differences in ecological product supply capacity, ecological compensation standards are determined and ecological compensation policies and measures are formulated so as to improve the ecological product supply capacity [93,94].” Is it about governmental lows or research?

Response 25: Thank you for your comments. In the process of ecological compensation, the government uses policy guidance to improve the enthusiasm of localities and residents, improve the efficiency of ecological protection, and enhance the willingness of residents to protect the ecological environment.

 Point 26: “...However, most of the existing research on improving the supply capacity of ecological products focuses on the social entities, and the research on the ecological factor endowment is insufficient.” What do you mean? There is a lot of studies about the effect of climate, soil etc. on ecosystem productivity, sustainability etc.

Response 26: Thank you for your comments. Although many studies have discussed the influencing factors of ecological assets/ecosystem services, these studies have not paid attention to how to improve the supply capacity of ecological products by adjusting the influencing factors of ecological assets, but only talk about ecological assets/ecological services.

 Point 27: “... village ecosystems...”do you mean the area of village itself (i.e., houses and adjacent small gardens) or more broad area including fields, meadows etc.?

Response 27: Thank you for your comment.

The village ecosystem is derived from the concept of "ecosystem", a natural, economic and socio-ecological network complex formed by human beings, resources and various environmental factors in village areas [120,127]. It consists of forest, grassland, water, farmland, garden, rural architecture and roads, and other landscapes, forming a unique landscape structure, livelihood structure and consumption structure through mechanisms such as material circulation, energy flow and information transfer, performing ecological functions, production functions and life functions. The village ecosystem is characterized by "health", "green" and "ecology" [128], with broad green open space, important air purification, climate regulation, water conservation and other ecological service functions, and is also an indispensable part of ensuring the high-quality and sustainable development of the region [129,130].

Round 2

Reviewer 1 Report

The revised manuscript has modified the related issues. I think it could accepted in present form.

Author Response

Thank you again for your comprehensive and constructive comments on our article, which helped us improve this manuscript.

Reviewer 2 Report

Congratulations to the authors on their attempts to improve the paper, which is now more coherent.  There are still issues with the English, including some very long sentences and some inappropriate words, but I will leave those issues for the editor to resolve. The paper is now more understandable for an English reader, and is scientifically sound. 

Author Response

(The authors gave the same response as above.)

Reviewer 3 Report

The quality of most parts of the manuscript considerably improved. However, the part 3.2.1 (the analysis of differences and common points between definitions of ecological assets, products and services in Tables 1-3) was not improved. In particular:

1.       In the Table 1 only Baiming Chen mention ownership in the definition of ecological assets. In contrast, Yaozhong Pan speaks about ecological assets at regional scale regardless of the ownership. Other mentioned authors did not mention ownership at all. However, in lines 204-206, 216-217 authors base the concept of ecological assets namely on the ownership. The authors gave the answer to this issue in their letter (Response 19), but I did not find this paragraph in the text. I think it should be included in a present form or somewhat changed.

2.       It would be good to give definitions of provision, support, regulation and cultural services

3.       The difference between ecological products and services is also not enough clear. In particular, ecological services (Table 2) in the definitions of Daily, De Groot, MA and Boyd are called “environmental conditions and processes…”, “Products and services…”, “benefits…” and “parts of the ecosystem…”, respectively, which means absolutely different things and must be discussed. According to lines 187-188 “In a narrow sense, ecological products are ecological services…”, so it is the same thing. According to the next sentence, “In a broad sense, ecological products … include not only the pure natural production of ecosystem services, but also the supply of agricultural and forestry products”. So, in this case ecosystem services are a subset of ecological products (i.e., all ecosystem services are ecological products, but not all ecological products are ecosystem services). If it is true, why the notion of “ecosystem services” is necessary at all? However, according to Fig.3, ecological products appear after determining ownership and market dealing, whereas ecosystem services directly follow from ecosystem and consequently are not a part of ecological products. In the lines 231-232 it is written that “…the number of tourists is used to characterize the supply capacity of ecological products”. Tourists are visiting certain ecosystem. So, what is product – ecosystem itself? And this ecological product is providing ecosystem service (cultural) - did I understand well?

4.       Also in Fig. 3 supply, regulation and cultural services appear twice: as ecosystem services (i.e., without determining ownership) and as ecological assets (i.e., after determining ownership). What is a difference? E.g., nice view (cultural service) is both, service and asset, if it is in a private area with entrance fee, but it is only ecosystem service if the access is free?

5.       So, make the order in these contradictions and explain the differences on simple examples!

6.       I would also replace the expression “rocky desertification control ecosystem” by something else (1) because it can be not a single type of ecosystems and (2) “control ecosystem” is usually used for comparison with an experimental ecosystem with some treatments applied. You can say, e.g. “ecosystems under rocky desertification control”, or “karst rocky desertification area”, or something like this.

7.       I think the term “village ecosystem” is not a good one. “It consists of forest, grassland, water, farmland, garden, rural architecture and roads, and other landscapes” (lines 539-540). These are several different ecosystems, and forest is usually not a part of a village. So, apparently under “village” you mention smallest administrative unit in a countryside or maybe the whole area owned by inhabitants of a particular village. This should be explained.

8.       Response 21: the text in the authors’ letter does not correspond to the text of manuscript (in manuscript it was not changed). The same for Response 24

Author Response

Response to Reviewer 3 Comments

Thank you very much for your comprehensive and constructive comments on our article again, which are helpful for us to further improve this manuscript. We fully accept your review comments, according to your relevant suggestions, we have made adjustments and changes in the corresponding parts of the newly submitted paper. We believe our response addresses all of the reviewer's concerns well. We hope that our revised manuscript will be accepted for publication. The specific modifications are as follows:

Point 1 : In the Table 1 only Baiming Chen mention ownership in the definition of ecological assets. In contrast, Yaozhong Pan speaks about ecological assets at regional scale regardless of the ownership. Other mentioned authors did not mention ownership at all. However, in lines 204-206, 216-217 authors base the concept of ecological assets namely on the ownership. The authors gave the answer to this issue in their letter (Response 19), but I did not find this paragraph in the text. I think it should be included in a present form or somewhat changed.

Response 1: Thank you for your comments. We apologize for the extra work you have added to our inattentive manuscript revision process. We made changes in manuscript. The detailed revisions are as follows: Scarcity and property rights are the social attributes of ecological assets. Ecological assets come from ecological resources. Ecological resources are not equal to ecological assets. Ecological resources can be transformed into ecological assets. The transformation of ecological resources into ecological assets requires certain conditions: scarcity, clear property rights, and the ability to generate income. Scarce ecological resources will stimulate people's desire to possess, and lead to changes in the owners of ecological resources rights and interests, thus transforming ecological resources into ecological assets. Therefore, ecological assets have scarcity and property rights.

Point 2: It would be good to give definitions of provision, support, regulation and cultural services

Response 2: Thank you for your sincere advice. We thank the reviewer for pointing out this issue. We supplement the definitions as follows: MA divides ecosystem services into provision services, support services, regulation services and cultural services. Provision services are the products obtained from ecosystems, including: food, water, wood and fibre etc.. Regulation services are the benefits obtained from the regulation of ecosystem processes, including the regulation of climate, floods, disease, waste, and water quality. Cultural services are the nonmaterial benefits people obtain from ecosystems through spiritual enrichment, cognitive development, reflection, recreation, and aesthetic experiences. Support services are those that are necessary for the production of all other ecosystem services. For example, production of atmospheric oxygen, soil formation and retention, nutrient cycling, water cycling, and provisioning of habitat.

Point 3: The difference between ecological products and services is also not enough clear. In particular, ecological services (Table 2) in the definitions of Daily, De Groot, MA and Boyd are called “environmental conditions and processes…”, “Products and services…”, “benefits…” and “parts of the ecosystem…”, respectively, which means absolutely different things and must be discussed. According to lines 187-188 “In a narrow sense, ecological products are ecological services…”, so it is the same thing. According to the next sentence, “In a broad sense, ecological products … include not only the pure natural production of ecosystem services, but also the supply of agricultural and forestry products”. So, in this case ecosystem services are a subset of ecological products (i.e., all ecosystem services are ecological products, but not all ecological products are ecosystem services). If it is true, why the notion of “ecosystem services” is necessary at all? However, according to Fig.3, ecological products appear after determining ownership and market dealing, whereas ecosystem services directly follow from ecosystem and consequently are not a part of ecological products. In the lines 231-232 it is written that “…the number of tourists is used to characterize the supply capacity of ecological products”. Tourists are visiting certain ecosystem. So, what is product – ecosystem itself? And this ecological product is providing ecosystem service (cultural) - did I understand well?

Response 3: Thank you for your sincere advice. We thank the reviewer for pointing out this issue. We have made changes in manuscript. We explain this issue as follows:

The difference between these definitions is whether to distinguish between the service generation process, the service acquisition process and the service final benefit. MA and Daily shows that as long as humans are the beneficiaries, ecosystem processes or functions are services. For Boyd services are directly consumed components (structure included), meaning indirect processes and functions are not ecosystem services. Wallace suggests that ecosystem processes should not be considered services. Ecosystem services are considered benefits by Costanza, whereas ecosystem services are not considered benefits by Fisher. Consistently, ecosystem patterns, processes and functions underlie the generation of ecosystem services. Ecosystem functions and ecosystem services do not correspond one-to-one. Ecosystem services and products are interdependent. Human well-being requires ecosystem services.

Whether defined in a broad or narrow sense, ecological products are part of ecosystem services. The reasons are as follows: Ecological products, in a narrow sense, are purely natural products and services produced by biological, such as wild fruits. In a broad sense, ecological products are supplemented with products of human labour, such as fruits grown by farmers. Agroforestry products with human labour attached can be understood as the provisioning services of ecosystems in which humans interact with nature.

 Ecological products can be divided into three types: material products, ecological regulation services, and ecosystem cultural services. Among them, cultural services include aesthetic viewing, leisure and entertainment, scientific research and education services provided by the ecosystem. Tourists can experience different cultural services including inspiration, cultural heritage, aesthetics, education, etc. in the process of leisure travel. The number of tourists indirectly represents the supply capacity of ecosystem cultural services. Tourists obtain cultural service products by visiting and viewing scenic spots.

Point 4: Also in Fig. 3 supply, regulation and cultural services appear twice: as ecosystem services (i.e., without determining ownership) and as ecological assets (i.e., after determining ownership). What is a difference? E.g., nice view (cultural service) is both, service and asset, if it is in a private area with entrance fee, but it is only ecosystem service if the access is free?

Response 4: Thank you for your sincere advice. We thank the reviewer for pointing out this issue. We made changes in manuscript. We explain this issue as follows:

We have made partial modifications to Figure 3. The reasons are as follows: Ecological assets have property rights. Ecosystem services as an ecological asset flow, ecosystem services also have property rights. Generally speaking, most of the ecological assets in physical form (such as fruits, timber, water resources, etc.) can determine the ownership. Invisible ecological assets (such as air purification and oxygen supply, climate regulation, landscape enjoyment, etc.) are mostly public, and it is difficult for any organization or individual to determine their ownership. When the ecological assets are clearly divided into ownership, they enter the market for trading, which can be converted into ecological products.

Point 5: So, make the order in these contradictions and explain the differences on simple examples!

Response 5: Thank you for your sincere advice. We thank the reviewer for pointing out this issue. We add relevant simple examples.

Point 6: I would also replace the expression “rocky desertification control ecosystem” by something else (1) because it can be not a single type of ecosystems and (2) “control ecosystem” is usually used for comparison with an experimental ecosystem with some treatments applied. You can say, e.g. “ecosystems under rocky desertification control”, or “karst rocky desertification area”, or something like this.

Response 6: Thank you for your pretty detailed comments. We carefully considered your comments and entirely agreed with you. We replaced rocky desertification control ecosystem to ecosystems under rocky desertification control.

Point 7: I think the term “village ecosystem” is not a good one. “It consists of forest, grassland, water, farmland, garden, rural architecture and roads, and other landscapes” (lines 539-540). These are several different ecosystems, and forest is usually not a part of a village. So, apparently under “village” you mention smallest administrative unit in a countryside or maybe the whole area owned by inhabitants of a particular village. This should be explained.

Response 7: Thank you for your comment. We thank the reviewer for pointing out this issue. We apologize for the extra work you have added to our inattentive language translation process. We made the following modifications and explanations.

It consists of landscapes such as forests, grasslands, water, farmland, gardens, rural buildings, and roads.

Our view is that we should delineate the boundary of the village ecosystem according to the watershed,  break the restrictions of administrative boundaries,and take the delineated range of the village ecosystem as the basic research unit. The reason as follows: The impact of some ecosystem services extends beyond the limits of administrative boundaries. A single village usually cannot constitute an independent ecological unit in terms of ecological environment. From the perspective of ecological environment unit, taking the watershed as the smallest basic regional unit can more scientifically and reasonably reflect the ecological environment of a region. The boundaries of the village ecosystems we delineated based on watersheds are no longer limited to the smallest administrative unit.

Point 8: Response 21: the text in the authors’ letter does not correspond to the text of manuscript (in manuscript it was not changed). The same for Response 24.

Response 8: Thank you for your comments. We apologize for the extra work you have added to our inattentive manuscript revision process. We have made changes in manuscript.

Response to Reviewer 3 Comments

Thank you very much for your comprehensive and constructive comments on our article again, which are helpful for us to further improve this manuscript. We fully accept your review comments, according to your relevant suggestions, we have made adjustments and changes in the corresponding parts of the newly submitted paper. We believe our response addresses all of the reviewer's concerns well. We hope that our revised manuscript will be accepted for publication. The specific modifications are as follows:

Point 1 : In the Table 1 only Baiming Chen mention ownership in the definition of ecological assets. In contrast, Yaozhong Pan speaks about ecological assets at regional scale regardless of the ownership. Other mentioned authors did not mention ownership at all. However, in lines 204-206, 216-217 authors base the concept of ecological assets namely on the ownership. The authors gave the answer to this issue in their letter (Response 19), but I did not find this paragraph in the text. I think it should be included in a present form or somewhat changed.

Response 1: Thank you for your comments. We apologize for the extra work you have added to our inattentive manuscript revision process. We made changes in manuscript. The detailed revisions are as follows: Scarcity and property rights are the social attributes of ecological assets. Ecological assets come from ecological resources. Ecological resources are not equal to ecological assets. Ecological resources can be transformed into ecological assets. The transformation of ecological resources into ecological assets requires certain conditions: scarcity, clear property rights, and the ability to generate income. Scarce ecological resources will stimulate people's desire to possess, and lead to changes in the owners of ecological resources rights and interests, thus transforming ecological resources into ecological assets. Therefore, ecological assets have scarcity and property rights.

Point 2: It would be good to give definitions of provision, support, regulation and cultural services

Response 2: Thank you for your sincere advice. We thank the reviewer for pointing out this issue. We supplement the definitions as follows: MA divides ecosystem services into provision services, support services, regulation services and cultural services. Provision services are the products obtained from ecosystems, including: food, water, wood and fibre etc.. Regulation services are the benefits obtained from the regulation of ecosystem processes, including the regulation of climate, floods, disease, waste, and water quality. Cultural services are the nonmaterial benefits people obtain from ecosystems through spiritual enrichment, cognitive development, reflection, recreation, and aesthetic experiences. Support services are those that are necessary for the production of all other ecosystem services. For example, production of atmospheric oxygen, soil formation and retention, nutrient cycling, water cycling, and provisioning of habitat.

Point 3: The difference between ecological products and services is also not enough clear. In particular, ecological services (Table 2) in the definitions of Daily, De Groot, MA and Boyd are called “environmental conditions and processes…”, “Products and services…”, “benefits…” and “parts of the ecosystem…”, respectively, which means absolutely different things and must be discussed. According to lines 187-188 “In a narrow sense, ecological products are ecological services…”, so it is the same thing. According to the next sentence, “In a broad sense, ecological products … include not only the pure natural production of ecosystem services, but also the supply of agricultural and forestry products”. So, in this case ecosystem services are a subset of ecological products (i.e., all ecosystem services are ecological products, but not all ecological products are ecosystem services). If it is true, why the notion of “ecosystem services” is necessary at all? However, according to Fig.3, ecological products appear after determining ownership and market dealing, whereas ecosystem services directly follow from ecosystem and consequently are not a part of ecological products. In the lines 231-232 it is written that “…the number of tourists is used to characterize the supply capacity of ecological products”. Tourists are visiting certain ecosystem. So, what is product – ecosystem itself? And this ecological product is providing ecosystem service (cultural) - did I understand well?

Response 3: Thank you for your sincere advice. We thank the reviewer for pointing out this issue. We have made changes in manuscript. We explain this issue as follows:

The difference between these definitions is whether to distinguish between the service generation process, the service acquisition process and the service final benefit. MA and Daily shows that as long as humans are the beneficiaries, ecosystem processes or functions are services. For Boyd services are directly consumed components (structure included), meaning indirect processes and functions are not ecosystem services. Wallace suggests that ecosystem processes should not be considered services. Ecosystem services are considered benefits by Costanza, whereas ecosystem services are not considered benefits by Fisher. Consistently, ecosystem patterns, processes and functions underlie the generation of ecosystem services. Ecosystem functions and ecosystem services do not correspond one-to-one. Ecosystem services and products are interdependent. Human well-being requires ecosystem services.

Whether defined in a broad or narrow sense, ecological products are part of ecosystem services. The reasons are as follows: Ecological products, in a narrow sense, are purely natural products and services produced by biological, such as wild fruits. In a broad sense, ecological products are supplemented with products of human labour, such as fruits grown by farmers. Agroforestry products with human labour attached can be understood as the provisioning services of ecosystems in which humans interact with nature.

 Ecological products can be divided into three types: material products, ecological regulation services, and ecosystem cultural services. Among them, cultural services include aesthetic viewing, leisure and entertainment, scientific research and education services provided by the ecosystem. Tourists can experience different cultural services including inspiration, cultural heritage, aesthetics, education, etc. in the process of leisure travel. The number of tourists indirectly represents the supply capacity of ecosystem cultural services. Tourists obtain cultural service products by visiting and viewing scenic spots.

Point 4: Also in Fig. 3 supply, regulation and cultural services appear twice: as ecosystem services (i.e., without determining ownership) and as ecological assets (i.e., after determining ownership). What is a difference? E.g., nice view (cultural service) is both, service and asset, if it is in a private area with entrance fee, but it is only ecosystem service if the access is free?

Response 4: Thank you for your sincere advice. We thank the reviewer for pointing out this issue. We made changes in manuscript. We explain this issue as follows:

We have made partial modifications to Figure 3. The reasons are as follows: Ecological assets have property rights. Ecosystem services as an ecological asset flow, ecosystem services also have property rights. Generally speaking, most of the ecological assets in physical form (such as fruits, timber, water resources, etc.) can determine the ownership. Invisible ecological assets (such as air purification and oxygen supply, climate regulation, landscape enjoyment, etc.) are mostly public, and it is difficult for any organization or individual to determine their ownership. When the ecological assets are clearly divided into ownership, they enter the market for trading, which can be converted into ecological products.

Point 5: So, make the order in these contradictions and explain the differences on simple examples!

Response 5: Thank you for your sincere advice. We thank the reviewer for pointing out this issue. We add relevant simple examples.

Point 6: I would also replace the expression “rocky desertification control ecosystem” by something else (1) because it can be not a single type of ecosystems and (2) “control ecosystem” is usually used for comparison with an experimental ecosystem with some treatments applied. You can say, e.g. “ecosystems under rocky desertification control”, or “karst rocky desertification area”, or something like this.

Response 6: Thank you for your pretty detailed comments. We carefully considered your comments and entirely agreed with you. We replaced rocky desertification control ecosystem to ecosystems under rocky desertification control.

Point 7: I think the term “village ecosystem” is not a good one. “It consists of forest, grassland, water, farmland, garden, rural architecture and roads, and other landscapes” (lines 539-540). These are several different ecosystems, and forest is usually not a part of a village. So, apparently under “village” you mention smallest administrative unit in a countryside or maybe the whole area owned by inhabitants of a particular village. This should be explained.

Response 7: Thank you for your comment. We thank the reviewer for pointing out this issue. We apologize for the extra work you have added to our inattentive language translation process. We made the following modifications and explanations.

It consists of landscapes such as forests, grasslands, water, farmland, gardens, rural buildings, and roads.

Our view is that we should delineate the boundary of the village ecosystem according to the watershed,  break the restrictions of administrative boundaries,and take the delineated range of the village ecosystem as the basic research unit. The reason as follows: The impact of some ecosystem services extends beyond the limits of administrative boundaries. A single village usually cannot constitute an independent ecological unit in terms of ecological environment. From the perspective of ecological environment unit, taking the watershed as the smallest basic regional unit can more scientifically and reasonably reflect the ecological environment of a region. The boundaries of the village ecosystems we delineated based on watersheds are no longer limited to the smallest administrative unit.

Point 8: Response 21: the text in the authors’ letter does not correspond to the text of manuscript (in manuscript it was not changed). The same for Response 24.

Response 8: Thank you for your comments. We apologize for the extra work you have added to our inattentive manuscript revision process. We have made changes in manuscript.
